# High expression of *Rex-orf-I* and *HBZ* mRNAs and bronchiectasis in lung of HTLV-1A/C infected macaques

Sarkis Sarkis [1]✉, Anna Gutowska [1], Mohammad Arif Rahman [1], Luca Schifanella[1], Katherine C. Goldfarbmuren [2,3], Massimiliano Bissa [1], Ramona Moles[1,4], Christina Ramirez[5], Elijah F. Edmondson [5], Andrew Warner [5], Melvin Doster[1], Isabela Silva de Castro[1], Robyn Washington-Parks[1], Sophia Brown[1], Joshua Kramer[6], Matthew W. Breed[5,7], Kristin E. Killoran [5], Yogita Jethmalani[8], Leonid Serebryannyy[8], Damian FJ. Purcell [9], Cynthia A. Pise-Masison[1,10] & Genoveffa Franchini [1]✉

HTLV-1 type-A rarely causes lung disease in humans, whereas HTLV-1 type-C is more frequently associated with respiratory failure and premature death. We investigated the genetic basis of HTLV-1C morbidity by constructing a chimeric HTLV-1A/C$_{ol-L}$ encompassing the highly divergent type C orf-I. We demonstrate that systemic infectivity of HTLV-1A and HTLV-1A/C$_{ol-L}$ is equivalent in macaques, but viral expression in lungs is significantly higher in HTLV-1A/C$_{ol-L}$ infection. In addition, bronchoalveolar-lavage immune cell dynamics differs greatly with neutrophils and monocytes producing TNF-α in HTLV-1A/C$_{ol-L}$, but producing IL-10 in HTLV-1A infection. Animals infected with HTLV-1A/C$_{ol-L}$ develops bronchiectasis at 10 months from infection, but at the same timepoint those infected with HTLV-1A do not. HTLV-1A/C$_{ol-L}$ expressed a 16 kDa fusion protein (p16C) via a doubly spliced, Rex-orf-IC, mRNA able to shield T-cells from efferocytosis, a monocyte function that mitigates inflammation via clearance of apoptotic cells. The Rex-orf-IC mRNA is expressed as more frequent in the lung of HTLV-1A/C$_{ol-L}$ than HTLV-1A infected animals. Since defective efferocytosis is associated with lung obstructive pathologies, the data raise the hypothesis that p16C may contribute to the lung morbidity observed in HTLV-1C infection.

Human T-cell leukemia virus type I (HTLV-1)[1] causes the neurodegenerative disease Tropical spastic paraparesis/HTLV-1-associated myelopathy (TSP/HAM) and inflammatory conditions such as dermatitis, arthritis, and uveitis[2–5]. HTLV-1 infection is endemic in regions worldwide[6–11] and despite the high genetic conservation of HTLVs, seven global subtypes have been identified based on the sequences of their viral promoters, long terminal repeats (LTRs), and envelopes[12–17].

HTLV-1A is the most common type globally. HTLV-1C, the most divergent variant[6,10,14,18], is endemic among Aboriginal populations in the Northern Territory, Australia, with a seroprevalence above 30%[19]. Clinical data from this population suggest a higher association of HTLV-1C infection with lung morbidity, possibly stemming from a genetic difference in the HTLV-1 subtypes[19–21]. The highest nucleotide divergence in the genomes of HTLV-1 A and C occurs in *orf-I*, encoded by a singly spliced *mRNA* in HTLV-1A, but lacking a translation initiation codon in HTLV-1C[14]. Prior work has demonstrated that *orf-I* expresses p12 and p8; proteins central to HTLV-1A persistence that increase STAT-5 signaling[22,23], downregulate the expression of MHC-I[24], ICAM-1,

and ICAM-2[25], and bind to the 16 kDa subunit of vacuolar ATPase[26]. The ER retention signal of p12 is cleaved to generate p8, which localizes to the cell surface and downregulates TCR signaling[27,28], induces conduit formation, and rapidly transmits virus to neighboring cells[29,30]. Indeed, p8 binds to vasodilator-stimulated phosphoprotein (VASP), which promotes actin filament elongation and facilitates Gag transfer via cellular conduits[31,32]. Genetic mutations at the p12 cleavage site affect the balance of p12/p8, which itself is associated with virus levels in infected individuals[27,33]. Both p12 and p8 are essential for viral infectivity, as ablation of *orf-I* expression or genetic mutations affecting the balance of p12/p8 impairs HTLV-1 infectivity in the macaque model[33–35].

While *orf-I* expression is central to HTLV-1A infectivity, early findings that HTLV-1C lacks the AUG initiation codon for *orf-I* translation[14] have recently been corroborated[10,11], leading us to hypothesize that HTLV-1C expresses *orf-I* by an alternative mechanism. To test our hypothesis, and since export of primary blood samples from Aboriginal people living with HTLV-1C to the United States is restricted, we constructed a chimeric HTLV-1A/C$_{ol-L}$ molecular clone in silico and demonstrated its infectivity in human and macaque CD4$^+$ T-cells in vitro and in macaques in vivo. We found that subtype C *orf-I* (*orf-IC*) can be expressed in vitro and in vivo in the lungs of infected macaques by a doubly spliced mRNA juxtaposed with the first exon of *rex*, which provides its ATG in frame with *orf-I* and encodes the p16C protein (*rex-orf-IC*). Although comparison of HTLV-1A and chimeric HTLV-1A/C$_{ol-L}$ viruses demonstrated similar systemic infectivity in macaques, the host inflammatory response in blood and bronchoalveolar lavage (BAL) differed significantly. Importantly, animals infected with HTLV-1A/C$_{ol-L}$ had higher levels of viral expression in the lung than those infected with HTLV-1A and developed interstitial pneumonia with infiltration by T-cells and B-cells, peripheral fibrosis, and bronchiectasis. Minimal lung viral expression and minimal lung pathology were observed in HTLV-1A-infected animals. In vitro, T-cells expressing p16C become resistant to engulfment by efferocytosis, a monocyte function that clears apoptotic cells and maintains tissue homeostasis. These data suggest the hypothesis that p16C expression in the lung of HTLV-1A/C infected animals may mediate the pathogenetic mechanism underlying the increased inflammation and lung disease, as efferocytosis is essential for lung homeostasis.

## Results

### In vitro infectivity of human and macaque CD4$^+$ T-cells in HTLV-1A/C$_{ol-L}$ infection

Expression of the *orf-I* encoded p12/p8 protein is essential for in vivo viral fitness in HTLV-1A infection[34,36]. To assess the role of *orf-I* in HTLV-1C, we constructed the chimeric HTLV-1A/C$_{ol-L}$ (*orf-I*-LTR) genome (Fig. 1a) by cloning the entire *orf-I*, the overlapping *orfs II, III, IV*, and the 3′LTR of HTLV-1C derived from an infected human patient[18] into the unique SalI-EcoRI site of the pAB HTLV-1A backbone (pAB_D26)[34,37] (see "Methods" and Supplementary Fig. 1a). Transfection of HTLV-1A/C$_{ol-L}$ and HTLV-1A plasmid DNAs into HEK293T cells demonstrated similar expression of Gag precursor, p24 protein, and the gp46 envelope protein (Fig. 1b). Expression of p40 Tax protein was detected only in the cell lysates of HTLV-1A but not HTLV-1A/C$_{ol-L}$, likely due to poor cross-reactivity of type A and C anti-Tax antibodies (Fig. 1b, Supplementary Fig. 1b, c). However, the level of p19 Gag matrix protein in the cell culture supernatant did not differ between viruses (Fig. 1c), suggesting comparable Tax activity on viral LTR. This notion was supported by a finding of equivalent Tax-A and Tax-C activity on their respective homologous or heterologous LTRs (Fig. 1d, e) as well as on the NF-κB promoter (Fig. 1f). We compared the infectivity of HTLV-1A and HTLV-1A/C$_{ol-L}$ by co-cultivation of infected γ-irradiated 729.6 B-cells with human umbilical cord blood cells from two individuals[34,38] and demonstrated equivalent virus production over time, measured as HTLV-1 p19 Gag production in the supernatant (Fig. 1g). Similarly, equivalent replication of the two viruses was observed using purified

CD4$^+$ T-cells from three healthy human donors (Fig. 1h) and three naïve juvenile rhesus macaques (Fig. 1i).

### HTLV-1A and HTLV-1A/C$_{ol-L}$ infection hastened by triple depletion of CD8$^+$ T-cells, NK cells, and monocytes

We recently established a robust animal model in which primary HTLV-1A infection is augmented by triple depletion of monocytes, CD8$^+$ T-cells, and NK cellular subsets[35,39]. Here, we performed two studies in macaques. In Study 1, we assessed the in vivo infectivity of the chimeric virus in a total of eight animals: four triple-depleted and four non-depleted macaques exposed intravenously to lethally γ-irradiated HTLV-1A/C$_{ol-L}$ 729.6 B-cells. Triple depletion was obtained by treating the animals with monoclonal antibody MT807R1 and Clodrosome®, or with IgG and Liposome as control (Fig. 2a, b). As expected, significant depletion of CD3$^+$CD8$^+$ and CD45$^+$NKG2A$^+$ cells was observed in triple-depleted animals (Supplementary Fig. 2a) at Day 0 (post-treatment and pre-inoculation) relative to baseline. At the same timepoint, absolute monocyte counts were significantly decreased in the triple-depleted group (Supplementary Fig. 2b). Interestingly, while the absolute counts of CD3$^+$CD4$^+$ decreased in the triple-depleted group, the frequency of CD3$^+$CD4$^+$ lymphocytes increased from baseline to Day 0 (Supplementary Fig. 2c). CD20$^+$ B-cell, neutrophil, and eosinophil absolute counts remained indistinguishable pre and post-treatment (Supplementary Fig. 2d). Following exposure to HTLV-1A/C$_{ol-L}$, three out of four triple-depleted animals seroconverted by week 3, and the fourth by week 5, with all animals remaining positive for antibodies to Gag (p24 and p19) and Env (rgp46-I and GD21) thereafter (Supplementary Fig. 3a, b). In contrast, only one of four replete animals (TiT) mounted a strong antibody response, but at a later time point (week 10; Supplementary Fig. 3c, d), and one did not seroconvert at all (16P042). The two remaining replete animals (TKM and DGM9) had low anti-p24 Gag antibodies, which declined over time, whereas antibodies to rgp46-I and GD21 were sustained (Supplementary Fig. 3c, d). As expected, viral DNA was found more frequently in the blood of triple-depleted animals than replete ones (Supplementary Fig. 3a, c, lower panels).

Study 2 was conducted at a later time point and consisted of six macaques simultaneously triple-depleted with the same reagents and regimen used in Study 1 and subsequently exposed at the same time points to lethally γ-irradiated 729.6 B-cells infected with HTLV-1A (n = 5) or HTLV-1A/C$_{ol-L}$ (n = 1; Supplementary Fig. 3e, f). Full seroconversion occurred in 5 of 6 animals between weeks 3–5 from exposure as expected and was sustained (Supplementary Fig 3e–h). One animal exposed to HTLV-1A did not fully seroconvert (TZW), but scored positive for viral DNA at weeks 16 and 21. As also observed in Study 1, viral DNA was detected in the other animals of this group, more frequently in blood and tissue (Supplementary Fig 3e, f, lower panel). The design of both studies is summarized in Supplementary Fig. 3i. This schematic also includes five historical animals exposed to HTLV-1A in identical conditions in a previous study, which were included in the subsequent analyses[39].

### Equivalent systemic virological parameters following exposure to HTLV-1A and HTLV-1A/C$_{ol-L}$

Next, we investigated HTLV-1A and HTLV-1A/C$_{ol-L}$ infectivity by calculating a virus score in vivo using the combination of number of bands observed on the nitrocellulose western blot strips, antibody titers, and virus distribution in blood over time in Studies 1 and 2 and compiled this data with that of the five additional historic animals exposed to HTLV-1A in identical triple-depletion protocol (Supplementary Table 1)[39] to calculate a composite systemic virus score accounting for all variables and timepoints (see "Methods"). Variables measured in Supplementary Fig. 3 were combined with the fraction of non-zero timepoints (Fig. 2c) to achieve a viral score for each timepoint (Fig. 2d). After performing similar calculations for the average variable score across all timepoints, we combined this variable-derived score with the

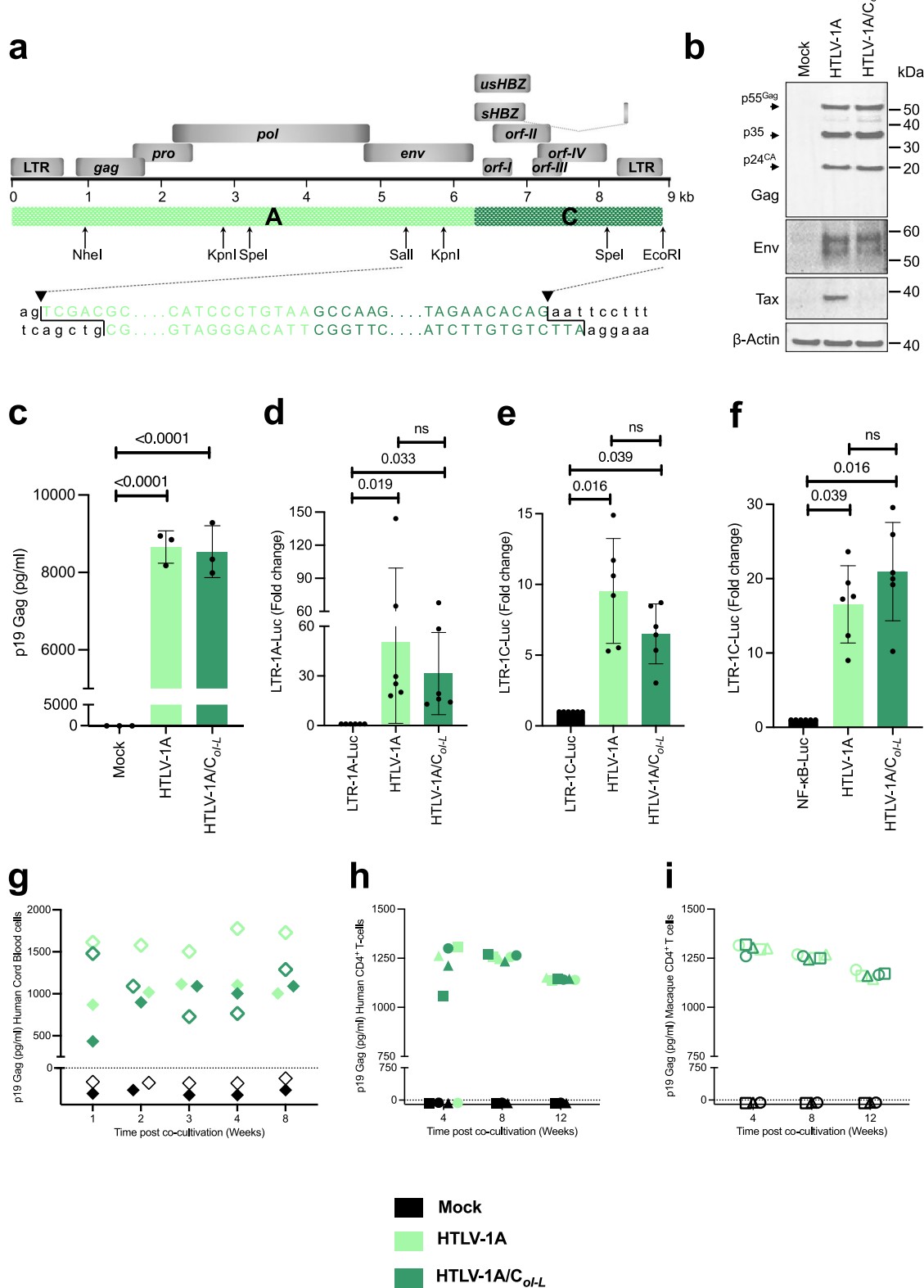

former timepoint-derived score to get the composite virus score (Fig. 2d). The systemic infectivity variables and viral scores did not significantly differ between HTLV-1A and HTLV-1A/C$_{ol-L}$ animals (Fig. 2c–e). As expected, however, many variables and scores were significantly lower in the non-depleted group (Fig. 2c–e). Overall, this analysis supports that virus replication following triple depletion is comparable for both HTLV-1A and HTLV-1A/C$_{ol-L}$.

## IL-8 and TNF-α, rather than IL-10, expressed preferentially by dendritic cells, monocytes, and neutrophils in HTLV-1A/C$_{ol-L}$ infection

Comparable kinetics of recovery in blood of CD8+ T-cells, NK cells, monocyte subsets, total lymphocytes (Supplementary Fig. 4a–c), CD20+ B-cells, neutrophils (Supplementary Fig. 4d–f), and CD4+ T-cell subsets (Supplementary Fig. 4g, h) were observed in the triple-

**Fig. 1 | HTLV-1A/C_{ol-L} chimeric molecular clone infectivity in humans and macaques in vitro. a** Genome organization of HTLV-1A (top) and schematic representation of HTLV-1A/C_{ol-L} chimeric virus (bottom). Restriction sites used in the construction and verification of the HTLV-1A/C_{ol-L} molecular clone are indicated with an arrow (see Supplementary Fig. 1a). Lowercase letters in black indicate the pAB_HTLV-1A backbone DNA sequence and uppercase letters indicate the inserted DNA fragment cassette including a portion of HTLV-1A envelope (light green) and the entire 3′ end region of the HTLV-1C spanning the *orf-I* to the 3′ *LTR* (dark green). **b** Representative western blot analysis of three independent experiment of Gag, Env, and Tax expression in HEK293T cells transiently transfected with HTLV-1A, HTLV-1A/C_{ol-L} molecular clones, or mock transfected. β-actin was used a loading control. **c** p19 Gag production in the cultures' supernatants of the 293 transiently transfected with HTLV-1A, HTLV-1A/C_{ol-L} molecular clones, or mock transfected were assessed by ELISA and presented as pg/ml. Data are mean ± SD from three biological replicates calculated by ordinary one-way ANOVA followed by Dunnet's

depleted groups exposed to either HTLV-1A or HTLV-1A/C_{ol-L}. Next, we performed in-depth phenotypical and functional analyses of dendritic cell, monocyte, and neutrophil frequencies in blood and in BAL (as a surrogate for lung) following viral exposure using the gating strategy described in (Supplementary Fig. 5a, b). Comparison of the impact of viral infection on blood and BAL dendritic cell subsets demonstrated that HTLV-1A/C_{ol-L} infection resulted in higher levels of myeloid dendritic cells (mDCs) and plasmacytoid dendritic cells (pDCs) producing IL-8 or TNF-α in blood compared to HTLV-1A (Fig. 3a, left and Supplementary Figs. 6a and 7a left, lower panels). While HTLV-1A/C_{ol-L} infected animals also initially recruited pDCs producing TNF-α and mDCs producing IL-8 (week 5) to the lung, the latter were subsequently reduced relative to those of HTLV-1A infected animals (Fig. 3a, right). In contrast, HTLV-1A/C_{ol-L} infection failed to recruit IL-8-producing pDC or TNF-α-producing mDCs to the lung at any time point compared to HTLV-1A-infected group (Fig. 3a, right). Consistently, however, HTLV-1A/C_{ol-L} infected animals had lower mDCs and pDCs producing IL-10 in both compartments (Fig. 3a). IL-8 and TNF-α are well-established key regulators of acute inflammatory response, recruitment and activation of monocytes and neutrophils to the site of inflammation. Accordingly, a higher frequency of all monocyte subsets producing TNF-α was observed in the blood of HTLV-1A/C_{ol-L} infected animals (Fig. 3b, left). While the latter group failed to recruit intermediate and non-classical monocytes producing TNF-α to the BAL during early infection (week 5), a higher frequency of TNF-α-producing classical monocytes, relative to HTLV-1A-infected animals, was detected at week 12 (Fig. 3b, right). HTLV-1A/C_{ol-L} infection resulted in lower frequencies of all monocyte subsets producing IL-10[+] in addition to those co-expressing CD162, over the course of the infection in both compartments (Fig. 3b and Supplementary Figs. 6b and 7b middle panels). Notably, IL-10[+] and CD162[+]IL-10[+] non-classical monocytes were an exception, displaying higher frequencies in the blood of HTLV-1A/C_{ol-L} infected animals at week 21 (Fig. 3b, bottom left and Supplementary Figs. 6b and 7b middle lower panel). Similarly, HTLV-1A/C_{ol-L} infection resulted in higher frequencies of pro-inflammatory markers of neutrophil activation in blood, such as IL-8, CD11b, and MPO (Fig. 3c, left and Supplementary Fig. 6c), markers associated with inflammatory diseases[40] and neutrophil-induced severe lung damage[41], as well as higher CD64[+] activated neutrophils and TNF-α producing neutrophils in BAL (Fig. 3c, right, and Supplementary Figs. 6c and 7c). In concert, IL-10 producing neutrophils had lower frequencies in HTLV-1A/C_{ol-L} infection (Fig. 3c). Collectively, these data demonstrated that while a predominance of myeloid cells and neutrophils producing pro-inflammatory cytokines was observed in both compartement of animals infected with HTLV-1A/C_{ol-L}, immune cells producing the anti-inflammatory, pro-resolution cytokine IL-10 predominated in both compartment of HTLV-1A infected animals.

multiple comparison test against mock with p = 0.00000073170 and 0.00000079517 for Mock vs. HTLV-1A and Mock vs. HTLV-1A/Col-L, respectively. Fold change induction of the luciferase activity of **d** LTR-1A, **e** LTR-1C, and **f** NF-κB. Data are mean ± SD from six biological replicates calculated by Kruskal–Wallis test followed by Dunn's multiple comparison test for group analysis. **d** ns > 0.999; **e**, **f** ns = 0.9645. p19 Gag production used as a surrogate marker for virus infection and virion production was assessed in the supernatant of the **g** human cord blood cells (n = 2, open and closed diamond shape), **h** human CD4[+] T-cells (n = 3, closed shapes), and **i** macaque CD4[+] T-cells (n = 3, open shapes) cocultured with γ-irradiated 729.6 B-cells producing either HTLV-1A or HTLV-1A/C_{ol-L} viruses. Each symbol represents **g** an independent human cord blood, **h** a healthy human donor, or **i** a juvenile naïve rhesus macaque. **c**–**i** Black, light green, and dark green correspond to the Mock, HTLV-1A, or HTLV-1A/Col-L viruses, respectively. Source data are provided as a Source data file.

## Differing spatiotemporal interactions of cytokines/chemokines and cell subsets in HTLV-1A and HTLV-1A/C_{ol-L} infection

To further investigate host responses to infection by HTLV-1A or HTLV-1A/C_{ol-L}, we conducted proteomic analysis of 45 cytokines/chemokines in the blood and BAL of both groups at weeks 5, 12, and 21 following virus exposure. We observed significant differences (Mann–Whitney, p < 0.05) between HTLV-1A/C_{ol-L} vs HTLV-1A in thirteen analytes in one or both compartments across different timepoints, confirming dynamic quantitative and qualitative differences in the systemic and mucosal immune responses elicited by each virus (Supplementary Fig. 8). Among these, the level of IL-6, a cytokine that promotes fibrosis by activating transforming growth factor beta (TGF-β)[42], was higher in blood at week 5 in HTLV-1A/C_{ol-L} infection. In BAL at 5 weeks from infection, the same group displayed elevated levels of the pro-inflammatory cytokine IL-1β, and the matrix metalloproteinase-1 (MMP1), which remodels tissues (Fig. 4a, Supplementary Fig. 8b). At week 12, the level of C-C chemokines ligand CCL2 (MCP-1) that recruits monocytes, of CCL13 (MCP-4), another chemotactic chemokine ligand that causes directional migration of monocytes, T-cells, eosinophil, and immature dendritic cells to inflammatory sites[43], and that of the IL-33 alarmin possibly released by damage tissues, were higher in the blood of HTLV-1A/C_{ol-L} infected animals. At the same timepoint, the levels of CCL3 (MIP-1α) and CCL19, chemokines that recruit monocytes, T-cells, and dendritic cells, were also elevated in the BAL of HTLV-1A/C_{ol-L} infected animals (Fig. 4b, Supplementary Fig. 8c). Notably, by week 21, IL-1β, IL-15, IL-18, IL-33, and FLT3LG (FMS-like tyrosine kinase 3 [*FLT3*] ligand LG), cytokine/chemokines that mobilize and cause differentiation and proliferation of dendritic cells, were elevated in the BAL of HTLV-1A infected animals (Fig. 4c, Supplementary Fig. 8d).

The impact of triple depletion on immune responses and cytokine profiles in both blood and bronchoalveolar lavage (BAL) fluid was compared between the replete (n = 4, HTLV-1A/C_{ol-L}) and triple-depleted (n = 5, HTLV-1A/C_{ol-L}; n = 10, HTLV-1A) animals at multiple timepoints (weeks 5, 12, and 21; Supplementary Figs. 4, 8 and 9). While significant increases in several cytokine/chemokine levels were observed in blood at an early timepoint in the triple-depleted animals (week 5 post-antibody treatment), as expected (Supplementary Figs. 8 and 9), these markers were comparable between triple-depleted and non-depleted animals by later timepoints (weeks 12 and 21; Supplementary Fig. 9a), after the recovery of key immune cell populations such as monocytes at week 1 and NK cells at week 5 (Supplementary Fig. 4a, b). In contrast, fewer significant changes in cytokine/chemokine levels between triple and non-depleted animals were detected in BAL at earlier stages (week 5), with the majority of differences, mostly consisting of decreased levels after triple depletion, observed only at later timepoints (weeks 12 and 21; Supplementary Fig. 9b). Importantly, most differences in BAL cytokine/chemokine levels between the two groups were unique to either the HTLV-1A/C_{ol-L} triple-depleted group

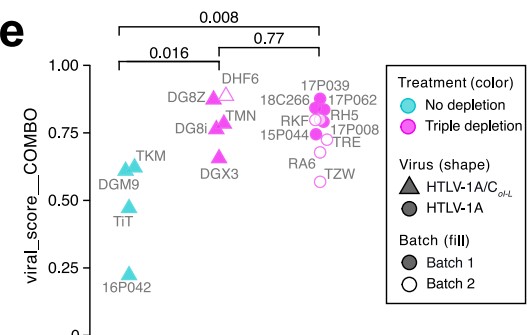

or the HTLV-1A triple-depleted group, relative to the HTLV-1A/C$_{ol-L}$ non-depleted group. This suggests that virus type prevails as the primary differentiating variable in the cytokine profiles.

Next, we addressed how significant differences in cytokine/chemokine levels between the two groups affected the frequency and function of cell subsets in blood and BAL. We performed correlation analyses, limiting our interpretations of the data to the strongest

Spearman correlations ($p < 0.01$). The low frequencies of IL-10 producing cell subsets observed in BAL during HTLV-1A/C$_{ol-L}$ infection were associated with elevated expression of MMP1 in BAL (for classical monocytes) and of IL-6 in blood (for neutrophils) at week 5 (Fig. 4d). Within the blood, elevated IL-6 levels also correlated with diminished frequencies of IL-10⁺ mDCs and pDCs as well as CD162⁺IL-10⁺ neutrophils (week 5; Fig. 4d), while elevated CCL2 correlated with depleted

**Fig. 2 | Virus score in triple-depleted animals exposed to HTLV-1A or HTLV-1A/C$_{oI-L}$. a, b** Schematic of the study design. **a** Black and **b** purple arrows correspond to the three consecutive days (Day −3,−2,−1) of treatment with either MT807R1 or IgG control antibodies, respectively, while **a** red and **b** orange arrows correspond to the treatments (Day −1) with either Clodrosome® or Liposome control. Dark green arrows in **a**, **b** indicate the inoculation day (Day 0) of the lethally γ-irradiated 729.6 B-cell line producing HTLV-1A/C$_{oI-L}$ virus. **c** Each heatmap block depicts the values for each viral variable (with from top to bottom: p24 titer, number of bands detected on a nitrocellulose strip, *gag* and *orf-I/II* detection by PCR) at each time-point (Weeks 3, 5, 7, 10, 12, 16, and 21) in each treatment (blue and pink)/virus (light and dark green) group, as well as the number of weeks not all zero for the variable as the top of each block. **d** Heatmap of the composite viral scores for each

timepoint, each variable, and combining, accompanying the methods for viral score generation. **c, d** Significance color bars at right of heatmaps designate whether the Mann−Whitney test (MW), for the pair-wise contrast at top is significantly up (red) /down (blue) or not significant (white) for the variable in each row. **e** Scatterplot of the 3 treatment/virus groups vs the final COMBO score (bottom line in the heatmap in **d**, Mann−Whitney p values for pair-wise comparisons among groups are shown. Closed triangles represent non-depleted (blue) or triple-depleted (pink) animals inoculated with HTLV-1A/C$_{oI-L}$ virus from study 1. Closed pink triangle represents the triple-depleted animal (DHF6) inoculated with HTLV-1A/C$_{oI-L}$ virus belonging to study 2. Closed and open pink circles represent triple-depleted animals inoculated with HTLV-1A virus from studies 1 and 2, respectively. Source data are provided as a Source data file.

IL-10$^+$ mDCs and CD162$^+$IL-10$^+$ classical or intermediate monocytes, and elevated IL-33 with lower IL-10$^+$ pDCs and intermediate monocyte frequencies (week 12; Fig. 4e). Depleted IL-10$^+$ mDCs in blood were also associated with elevated BAL levels of IL-1β (week 5) and CCL19 (week 12), (Fig. 4d, e, respectively). Together, these associations suggest that these cytokines/chemokines work in concert to decrease the circulation and migration of anti-inflammatory cells to the lung. Additionally, the elevated frequency of monocytes, DCs, or neutrophil subsets producing inflammatory markers IL-8, TNF-α, CD11b, MPO, or CD64 in both compartments in HTLV-1A/C$_{oI-L}$ infection correlated directly with increased levels of BAL MMP1, blood IL-6, (week 5; Fig. 4d), BAL CCL3, or CCL19 (week 12; Fig. 4e), and decreased levels of BAL IL-18 (week 21; Fig. 4f). At this later timepoint in HTLV-1A/C$_{oI-L}$ infection, lower BAL levels of IL-1β and IL-15 correlated with decreased frequencies of BAL mDC producing IL-8 (Fig. 4f), while the lower BAL levels of IL-18 and IL-15 associated with lower frequencies of systemic MPO producing neutrophils and higher frequencies of IL-10 producing non-classical monocytes (Fig. 4f). Further, these lower levels of IL-18 in HTLV-1A/C$_{oI-L}$ infection also correlated with decreased IL-10 producing monocyte subsets in BAL, IL-10$^+$ mDCs in both compartments and TNF-α$^+$ neutrophils in blood, and with elevated frequencies of blood neutrophils producing IL-8 or CD11b (Fig. 4f). Together, these data suggest that HTLV-1A/C$_{oI-L}$ infection may initially (weeks 5 and 12) orchestrate a pro-inflammatory immune environment in both compartments (Fig. 4g, top right) while consistently failing to mobilize pro-resolution IL-10 producing immune subsets systemically and ultimately in the lung (Fig. 4g, left).

## Bronchiectasis in HTLV-1A/C$_{oI-L}$ but not HTLV-1A infection

As demonstrated above, HTLV-1A/C$_{oI-L}$ infection in Study 1 was associated with complex temporospatial interactions of immune cells with pro-inflammatory cytokines/chemokines. To investigate whether HTLV-1A and /or HTLV-1A/C$_{oI-L}$ infection induced lung injuries, two triple-depleted macaques (TMN and DG8Z), one replete animal (TiT) infected with HTLV-1A/C$_{oI-L}$ in Study 1, and three HTLV-1A-infected animals in Study 2 (RFK, RH5, TRE), that had equivalent systemic virus scores, were sacrificed (week 48). The entire lungs of the animals (n = 6) were harvested, and histopathological analysis was performed on all seven lung lobes from each animal. In the three animals infected with HTLV-1A/C$_{oI-L}$, we observed interstitial pneumonia, alveolitis, and prominent fibrosis in animal TMN, and bronchiectasis in both the replete animal TiT and the triple-depleted animal DG8Z (Fig. 5a and Supplementary Table 2a). Less severe lesions were also observed in these animals, such as pulmonary hemorrhage with red blood cell presence within the alveolar spaces (Supplementary Table 2a). In the three HTLV-1A-infected animals, we observed mild to moderate fibrosis but no bronchiectasis (Supplementary Table 2b). RNAscope was then used to assess the presence of virus expression in regions overlapping the lung lesions in two of the HTLV-1A/C$_{oI-L}$- infected animals. A total of seven lung lobes from the two infected animals, plus an additional uninfected macaque as control, were stained with HTLV-1 *gag* probe (Supplementary Table 4). RNAscope revealed *gag* RNA in

samples from the lungs of the two HTLV-1A/C$_{oI-L}$ infected animals, TMN, and TiT (Fig. 5b, left and right, respectively), with rare positive cells morphologically consistent with infiltrating mononuclear immune cells. *Mmu-PPIB* and *dapB* were respectively used as positive and negative RNAscope controls to stain the left cranial lobe of animal DG8Z (Supplementary Fig. 10a, b). Next, we performed immune-histopathology on tissue samples that scored positive for viral expression by RNAscope and found multifocal regions with infiltrating immune cells, often surrounding small to medium caliber vessels. Phenotypic characterization of spatial immune inflammatory infiltration in lung tissue from all three animals infected with HTLV-1A/C$_{oI-L}$, demonstrated accumulation of CD20$^+$ B-cells interspersed with CD3$^+$ T-cells as well as Iba1$^+$ monocytes/macrophages (Fig. 5c and Supplementary Fig. 10c). Expression of smooth muscle actin (SMA), a well-established biomarker of activated fibroblasts tightly associated with lung disorders and fibrosis, was found to be highly expressed in the lung fibrotic sections of the three animals (Fig. 5c and Supplementary Fig. 10c). Together, the data document active HTLV-1 expression of the structural *gag* gene in lung lesions, and accumulation of immune cells surrounding the lesion area in HTLV-1A/C$_{oI-L}$, infected animals.

## HTLV-1C *orf-I* is expressed by doubly spliced *rex-orf-I* mRNA encoding p16C

The highest genetic divergence between HTLV-1A and C resides in *orf-I*. While in HTLV-1A, *orf-I* is expressed by a singly spliced mRNA that carries the initiating codon for translation[44–46], all HTLV-1C genotypes analyzed so far lack the translation initiation codon for *orf-I*[8,47]. We analyzed viral mRNA expression in HTLV-1A/C$_{oI-L}$ infection and compared it to 729.6 B-cells infected with HTLV-1A using specific type A and type C primers (Supplementary Fig. 11a and Supplementary Table 5). The four splice acceptor sites located in the pX region at positions 6383, 6478, 6881, and 6956 previously described in HTLV-1A were conserved between type A and type C variants[44] (Supplementary Fig. 11a, b). HTLV-1A/C$_{oI-L}$ expressed all expected un-spliced, singly spliced, and doubly spliced viral messenger RNAs encoding known HTLV-1 structural and regulatory proteins (Supplementary Fig. 11b). In in vitro HTLV-1A/C$_{oI-L}$ 729.6 infected B-cells, we observed low expression of the *rex-orf-I* mRNA that juxtaposes the first exon of *rex* in frame with *orf-I* and could encode a fusion protein of 152 amino acids translated from the AUG Rex initiation codon (Supplementary Fig. 11c). Expression of the *rex-orf-I* mRNA was further confirmed using the HTLV-1A/C$_{E-L}$ chimeric virus that was generated with a portion of *pol* gene and the entire *envelope* and *3′ end* of the HTLV-1C (HTLV-1A/C$_{E-L}$; Supplementary Fig. 11d–f).

## p16C, encoded by *rex-orf-I*, doubly spliced mRNA inhibits efferocytosis

Next, we synthesized in silico the cDNA of both HTLV-1A and HTLV-1C, adding the *rex-orf-I* mRNAs to each HA tag[22,48,49]. We demonstrated that both cDNAs express equivalent amounts of 16 kDa proteins (p16A and p16C) in Hela cells (Supplementary Fig. 12a). The HTLV-1A cDNA expressing the singly spliced HA tagged *orf-I mRNA* was used as

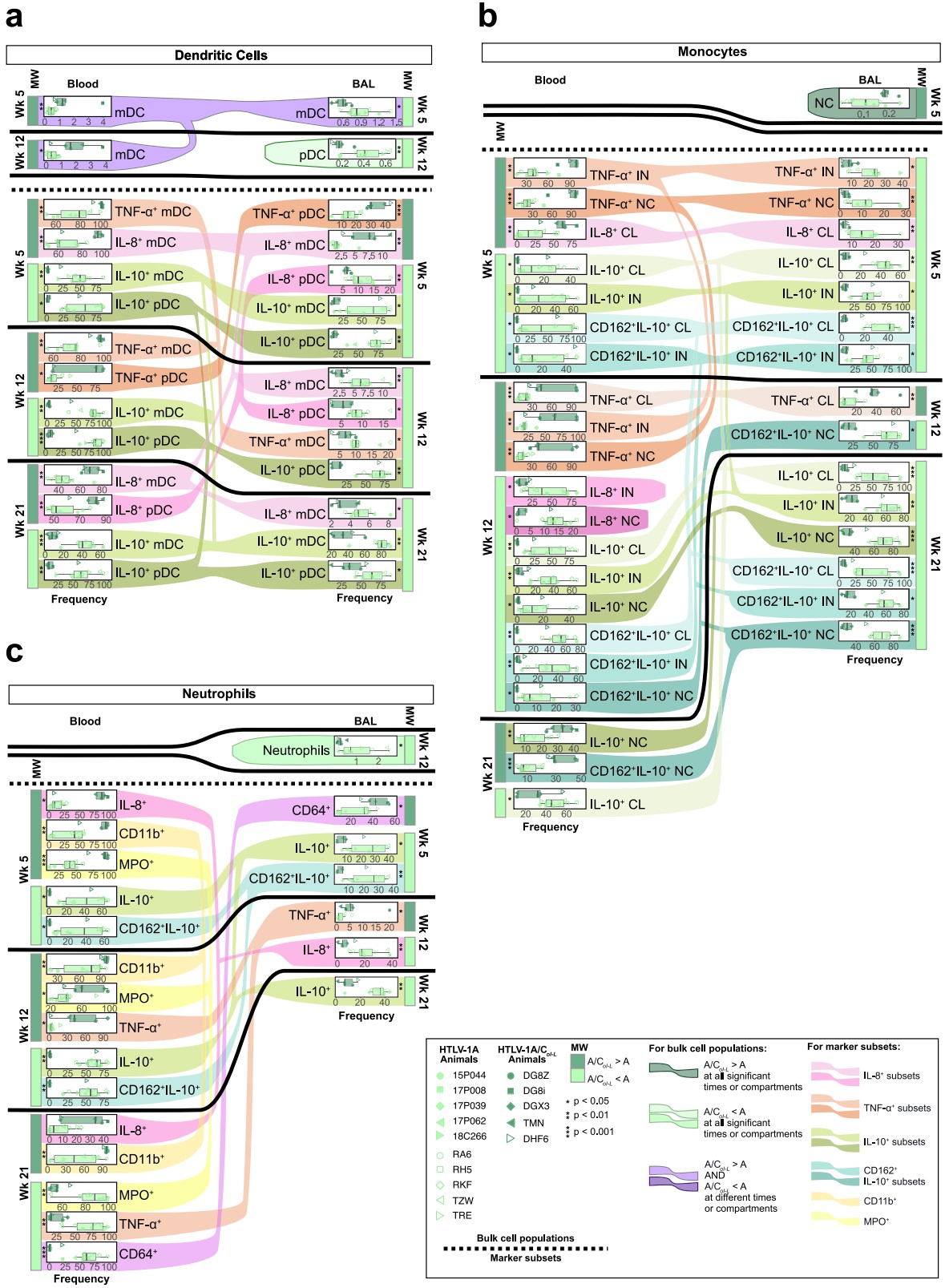

positive control and expressed p12/p8 as expected (Supplementary Fig. 12a). While expression of *rex-orf-I* mRNA following stable transduction of THP-1 or Jurkat T-cells yielded equivalent levels of mRNA for both viruses (Supplementary Fig. 12b), surprisingly p16C but not p16A was detected in the Jurkat T-cell line (Supplementary Fig. 12c). In THP-1 cells, the singly spliced *orf-I* encoded p12 precursor protein was completely cleaved to yield p8, as expected[33], and although p16A was

barely detectable, p16C expression was high and yielded additional discrete bands, suggesting cleavage (Supplementary Fig. 12d). The localization of p12/p8, p16A, and p16C in HeLa cells was mostly cytoplasmic as expected (Supplementary Fig. 12e–g), and both p16A and p16C localized to the ER (Supplementary Fig. 12h).

Since defective efferocytosis is a hallmark of lung disease[50], we hypothesized the HTLV-1C encoded p16C may affect this IL-10

**Fig. 3 | Neutrophil, monocyte, and dendritic cell dynamics in blood and BAL of HTLV-1A and HTLV-1A/C$_{ol-L}$ infected macaques.** Beeswarm boxplots depict cell population frequencies of **a** plasmacytoid dendritic cells (pDCs) and myeloid dendritic cells (mDCs) and their subsets expressing IL-8, IL-10, or TNF-α markers. **b** Total monocytes, with the three monocyte populations (CD14$^+$CD16$^-$ classical, CL; CD14$^-$CD16$^+$ non-classical, NC; and CD14$^+$CD16$^+$ intermediate, IN), and their subsets expressing IL-8, IL-10, both CD162 and IL-10, or TNF-α markers. **c** Neutrophils and their subsets expressing IL-8, IL-10, both CD162 and IL-10, TNF-α, CD11b, CD64, or MPO markers in blood (left) and BAL (right). The x-axis represents cell frequency. Only cell populations with p < 0.05 by Mann–Whitney test between the two triple-depleted groups at weeks (Wk) 5, 12, or 21 are shown and sorted by timepoint (top to bottom, separated by thick black lines), then by direction of the HTLV-1A/C$_{ol-L}$ vs HTLV-1A difference. P value ranges are indicated by asterisks: *p < 0.05, **p < 0.01, ***p < 0.001. Alluvials connect the same cell populations in the blood and BAL over time. Bulk population and subset frequencies are displayed above and below the dotted line, respectively. For bulk populations, alluvials are colored according to their pattern of significance with consistently HTLV-1A/C$_{ol-L}$ > HTLV-1A (dark green), or HTLV-1A/C$_{ol-L}$ < HTLV-1A (light green) at all significant timepoints/compartments. Purple shading indicates HTLV-1A/C$_{ol-L}$ greater than HTLV-1A, and HTLV-1A/C$_{ol-L}$ less than HTLV-1A at different timepoints/compartments. For subsets, alluvials are colored by marker cytokines as indicated. Alternative visualizations of this data can be found in Supplementary Figs. 6 and 7. While each dark and light green symbol corresponds to an animal from the triple-depleted groups inoculated with HTLV-1A/C$_{ol-L}$ or HTLV-1A, respectively, closed and open symbols correspond to animals belonging either Study 1 or Study 2, respectively. For box and whiskers plots, the center is the median (thick gray bar), bounds of the boxplot are the 25th and 75th percentiles, and whiskers extend from these bounds to the maximum and minimum value no further than 1.5 times the distance between the first and third quartiles. Source data are provided as a Source data file.

dependent monocyte function. We used primary human CD14$^+$ monocytes and the THP-1 cell line as effector cells to assess their ability to engulf apoptotic Jurkat T-cells constitutively expressing or not p16C, and p8 as control, in an efferocytosis assay. We found that p16C expression in Jurkat T-cells markedly decreased their susceptibility to engulfment by both human primary CD14$^+$ monocytes and THP-1 cells (Supplementary Fig. 13a, b). In contrast, in the same experiments, Jurkat T-cells constitutively expressing the HTLV-1A p8, encoded by the singly spliced *orf-I mRNA*, did not impair efferocytosis mediated by CD14$^+$ primary monocytes or THP-1 cells (Supplementary Fig. 13a, b). To begin addressing p16C function mechanistically, we measured the level of IL-10, a cytokine that promotes monocyte efferocytosis, in the supernatant of THP-1 monocytic cell lines constitutively expressing the HTLV-1A p8 or HTLV-1C p16C proteins (Supplementary Fig. 13a, b). The level of IL-10 was lower in the supernatant of THP-1 cells expressing p16C than in THP-1 expression p8 (Supplementary Fig. 13c). This is consistent with our in vivo observations of decreased IL-10 production by monocytes, DCs, and neutrophils in HTLV-1A/C$_{ol-L}$ infected animals (Fig. 3, Supplementary Figs. 6 and 7).

## Higher lung virus burden in HTLV-1A/C$_{ol-L}$ than in HTLV-1A-infected macaques

Next, we explored viral DNA distribution and viral mRNA expression in the right cranial, middle, and caudal lobes, the left cranial/cranial, cranial/caudal, and caudal lobes, and in the single accessory lobes of HTLV-1A and HTLV-1A/C$_{ol-L}$ infected animals (Fig. 6a). The entire lungs were collected at time of euthanasia (48 weeks post viral inoculation; Supplementary Fig. 3i) from four macaques infected with HTLV-1A/C$_{ol-L}$, (Study 1, TiT, TMN, DG8Z; Study 2, DHF6) and all five macaques infected with HTLV-1A (Study 2). Gag and orf-I/II DNAs and *gag*, *rex-orf-I*, *tax*, *usHBZ*, and *sHBZ mRNAs* were detected by PCR and RT-PCR, respectively (Supplementary Table 3a, b). The frequency of Gag DNA detection across all 7 lobes did not differ between the two groups, whereas that of *orf-I/II* viral DNA was higher in HTLV-1A/C$_{ol-L}$ infected animals (Fig. 6b). At the RNA level, *gag*, *rex-orf-I*, and spliced HBZ RNAs were more frequently detected in the lungs of HTLV-1A/C$_{ol-L}$ infected animals (Fig. 6c). Collective analysis of histological (Fig. 6d and Supplementary Table 2) and the virological parameters described above (Fig. 6b, c and Supplementary Table 3a, b) demonstrated a trend toward higher frequency of positive variables in the right cranial, right and left caudal, and accessory lobes of HTLV-1A/C$_{ol-L}$ infected animals (Fig. 6e). The presence of virus was documented in all 7 lobes of all three HTLV-1A/C$_{ol-L}$ infected animals. However, virus was detected in only 1 to 4 lobes in animals infected with HTLV-1A (Fig. 6e and Supplementary Table 3a, b). Altogether, these data demonstrate that while the systemic virus score did not differ in animals infected by the two viruses, viral genomic DNA and *gag*, *HBZ*, and *rex-orf-I mRNAs* expression was detected with higher frequency in the lung of HTLV-1A/C$_{ol-L}$ than in HTLV-1A.

## Discussion

HTLV-1C is associated with a high rate of early mortality attributed to lung inflammation and bronchiectasis in the Aboriginal communities of Australia, where HTLV-1C is endemic[19–21]. In sharp contrast, HTLV-1A causes mostly subclinical respiratory manifestations in a small portion of infected patients[51–53]. The viruses differ most in *orf-I*, with HTLV-1C lacking entirely the *orf-I* translation initiation codon that is indispensable to HTLV-1A infectivity[14]. In HTLV-1A, the *orf-I* translation initiation codon is necessary for expression of the singly spliced *mRNA* for the p12 precursor protein, cleaved to p8[27,28]. These proteins from HTLV-1A counteract adaptive and innate host immunity[23–25,31,32,35,54] and play a central role in HTLV-1 infectivity in vivo, but not in vitro. We hypothesized that HTLV-1C may express *orf-I* by alternative means, and since *orf-I* is the most genetically divergent viral gene, we investigated its transcription. We found that in HTLV-1A/C$_{ol-L}$, a doubly spliced mRNA (*rex-orf-I* mRNA) that juxtaposes the first exon of Rex in frame with *orf-I* encodes a 16 kDa protein (p16C), able to inhibit efferocytosis.

We confirmed here that simultaneous, triple depletion of monocytes, NK cells, and CD8$^+$ T-cells establishes a robust and persistent infection of animals with HTLV-1A, as observed previously[39], and extended this observation to HTLV-1A/C$_{ol-L}$ infection. Although we observed no differences in systemic dissemination of either virus, HTLV-1A/C$_{ol-L}$ infection was associated with higher virus burden in the lung. Interestingly, functional analysis of monocytes, DCs, neutrophils, and cytokine/chemokine levels demonstrated strikingly different spatiotemporal dynamics in blood and BAL following infection. Infection with HTLV-1A/C$_{ol-L}$ resulted in low overall frequency of monocytes, DCs, and neutrophils producing IL-10, with elevated frequencies of those producing TNF-α in both compartments, linked to high expression of IL-6, CCL2, and IL-33 in blood and of MMP1, IL-1β, CCL3, and CCL19 in the lung.

In the BAL of the HTLV-1A infected group at week 21, the elevated levels of IL-18, IL-15, and IL-1β were linked to high frequencies of mDCs, classical and non-classical monocytes producing IL-10, and mDCs producing IL-8, along with MPO or TNF-α producing neutrophils and IL-10 producing mDC in blood. While the depleted levels of IL-18 and IL-15 in the BAL of HTLV-1A/C$_{ol-L}$ infection favor a different mix of systemic pro-inflammatory and pro-resolution subsets, defined by elevated neutrophils producing IL-8 and CD11b alongside IL-10-producing non-classical monocytes, HTLV-1A/C$_{ol-L}$ infection again fails to recruit pro-resolution (IL-10) immune subsets to the lung at this late timepoint. Together, these data suggest that HTLV-1A/C$_{ol-L}$ infection results in higher and persistent inflammation at the expense of IL-10-mediated pro-resolution myeloid populations in blood and BAL compared with HTLV-1A.

Importantly, the distinct immune profile between HTLV-1A and HTLV-1A/C$_{ol-L}$ infection observed in this study leveraged animals infected with HTLV-1A in the present work and a prior independent study[39], supporting different engagement of host immunity by the two

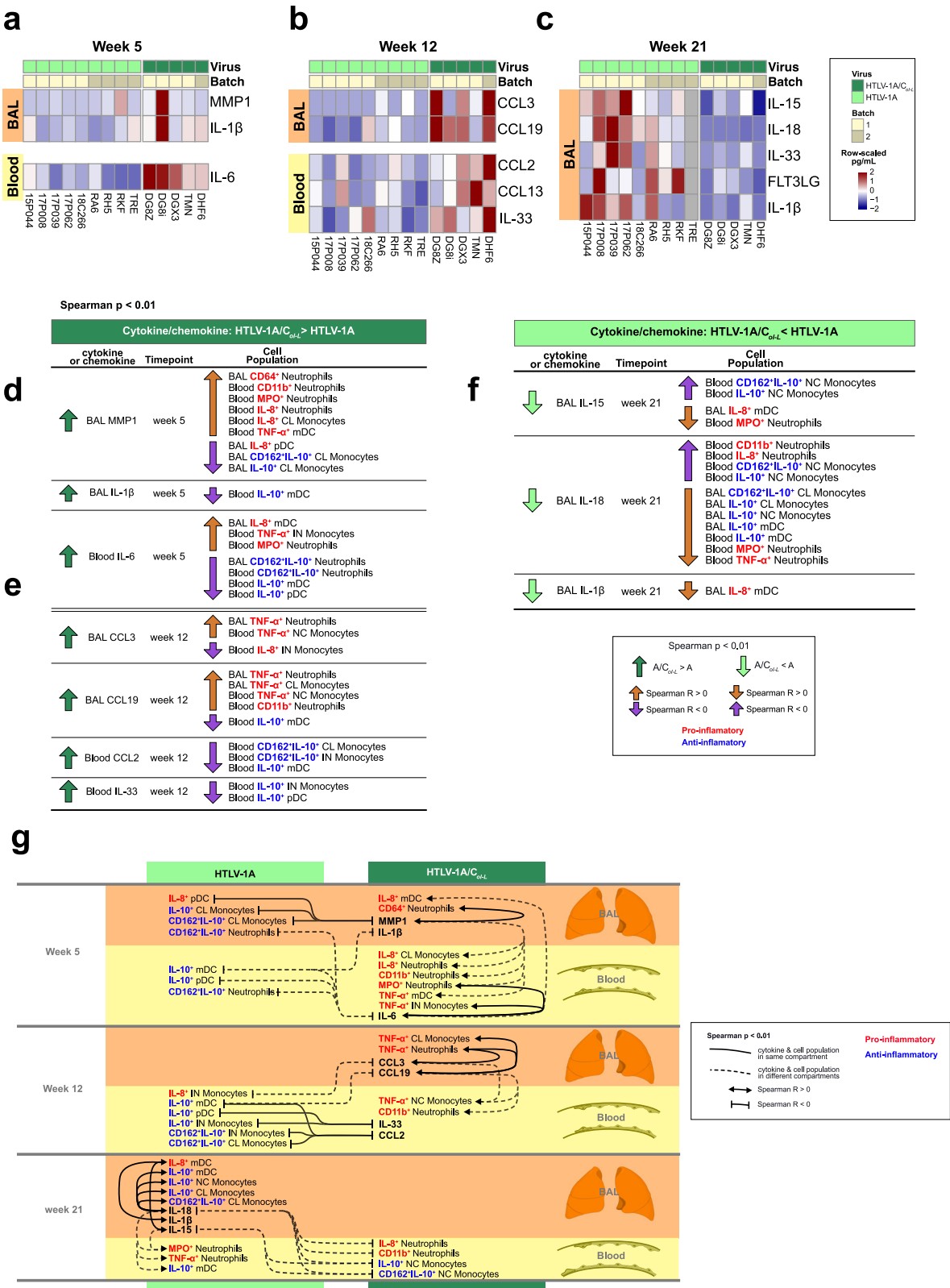

viruses across multiple animal cohorts. Defective efferocytosis results in incomplete clearance of apoptotic cells and tissue inflammation and plays a central role in pulmonary fibrosis[50,55–57]. We found that *orf-I* in HTLV-1A/C*ol-L* is expressed via a doubly spliced *rex-orf-I* mRNA and encodes a protein of 16kD that inhibits efferocytosis. This finding raises the hypothesis that the higher inflammatory profile, lacking IL-10-producing innate immune cells, and lung bronchiectasis of HTLV-1A/

C*ol-L* may be linked to defective clearance of apoptotic cells. Consistent with this hypothesis, expression of p16C in the THP-1 cell lines decreases IL-10 expression, a cytokine that promotes efferocytosis in monocytes.

In HTLV-1A, the presence of *orf-I* expression (relative to HTLV-1A_{p12KO}) results in upregulation of the do not eat me CD47 marker, which also renders infected T-cells less susceptible to monocyte

**Fig. 4 | Distinct cytokine and chemokine profiles in blood and BAL of HTLV-1A/C$_{oI-L}$ and HTLV-1A infected macaques and their correlations with cell populations in blood and BAL.** Heatmaps depict row-scaled pg/mL of cytokines/chemokines that are significantly different (Mann–Whitney, p < 0.05) between HTLV-1A/C$_{oI-L}$ and HTLV-1A at the timepoint indicated in **a**, **b** blood and BAL, and **c** BAL. **d–f** Cytokines/chemokines that differed between HTLV-1A/C$_{oI-L}$ and HTLV-1A in **a–c** that significantly correlated (Spearman p < 0.01) with significantly different cell populations at weeks 5, 12, and 21 post viral inoculation. Cytokines/chemokines are organized based on their comparative frequency at the time point and compartment indicated. Cytokines/chemokines higher in HTLV-1A/C$_{oI-L}$ than HTLV-1A (**d**, **e**) are indicated by upward, dark green arrows, and those lower in HTLV-1A/C$_{oI-L}$ (**f**) are shown with downward, light green arrows. In each row, the frequency of cell populations relative to the associated cytokine/chemokine is indicated by orange and purple arrows corresponding to positive (Spearman correlation R > 0) or negative correlations (R < 0), respectively. Pro-inflammatory cell population markers are highlighted in red, while anti-inflammatory cell population markers are highlighted in blue. **g** Schematic of Spearman associations with p < 0.01 between the cytokines in **a–c** and the cell populations in Fig. 3 at 5 (top), 12 (middle), or 21 (bottom) weeks post viral inoculation. For each time point, significantly associated cytokines or cell populations that are higher in HLTV-1A are on the left (light green column), and those higher in HTLV-1A/C$_{oI-L}$ are on the right (dark green column). Cytokines or cell populations found in the BAL are above with the orange underlay, while those in the blood are below with the yellow underlay. Solid lines connect associated cytokines/cell populations that are in the same compartment, while dashed lines connect cytokine/cell populations in different compartments. Positive associations are designated by arrows on the outer columns of the figure, while negative associations are designated by inhibitory vertical bars in the central region of the figure. Pro-inflammatory cell population markers are highlighted in red, and anti-inflammatory cell population markers are highlighted in blue. Lung scheme was created in BioRender. Sarkis, S. (2025) https://BioRender.com/4uohazj. Source data are provided as a Source data file.

engulfment[35]. We show here that overexpression of the HTLV-1A encoded *orf-I* p12 protein in isolation is not sufficient to inhibit efferocytosis in vitro, suggesting that additional viral genes may be necessary for CD47 upregulation observed in HTLV-1A infection[35,41]. While other viral genes may also be involved in the HTLV-1C inhibition of efferocytosis, p16C expression alone was sufficient to reduce efferocytosis in vitro. Thus, while both HTLV-1-A and HTLV-1C "target" efferocytosis, a basic mechanism of immune defense against invading pathogens, HTLV-1C appears to do so more efficiently. The mechanism by which p16C inhibits efferocytosis is not understood at present, and our data do not directly prove it is pathogenic in the pulmonary lesions observed in HTLV-1A/C infection. However, the data reported here establish a macaque model where this hypothesis can be further investigated.

Here, we demonstrate bronchiectasis and increased viral burden in the lung of animals infected with HTLV-1C and mild/moderate fibrosis in HTLV-1A infection in a relatively short time (~10 months from infection), supporting a causal role for HTLV-1C in lung disease, as observed in an endemic area in Australia[19–21]. HTLV-1A has also been shown to rarely cause lung disease, and our data demonstrating biological differences between the inflammatory profiles induced by HTLV-1A and a chimeric virus carrying the 3′ end of HTLV-1C provides a possible explanation for the differential lung morbidity associated with infection by HTLV-1A and HTLV-1C.

Importantly, the demonstration of active viral expression in lung of HTLV-1A/C$_{oI-L}$ infected animals suggests that antiviral treatment, possibly early in infection, may be beneficial to curb lung morbidity. This is particularly important since there are only two FDA-approved drugs that decrease proliferation and differentiation of lung fibroblasts[58], and they represent a significant economic burden. Limitations of our study include the small number of animals studied, and future work will be needed to confirm the causative role of HTLV-1C in pulmonary morbidity and the effect of p16C expression in lung damage, possibly by using a molecular clone of HTLV-1A/C knocked out for *rex-orf-I* expression. Despite these limitations, the availability of a relevant animal model of HTLV-1C infection may allow the testing of approaches to target viral replication and p16C function and effectively prevent HTLV-1C-associated lung morbidity.

## Methods

### Ethics statement
The Indian rhesus macaques (*Macaca mulatta*) used in this study were obtained from the free-range colony on Morgan Island (South Carolina) or Covance Research Products (Princeton, NJ) (Supplementary Table 1). Animals were housed and maintained at the NCI Animal Facility at the NIH, Bethesda, MD. The NIH is accredited by AAALAC International and follows the Public Health Service Policy for the Care and Use of Laboratory Animals. Animal care was provided in accordance with the procedures outlined in the "Guide for Care and Use of Laboratory Animals" (National Research Council, 2011; National Academy Press, Washington, D.C.). Animals were handled in accordance with AAALAC standards in an AAALAC-accredited facility (OLAW, Animal Welfare Assurance A4149-01 for NIH). All animal care and procedures were carried out under protocols approved by the NCI and/or NIAID Animal Care and Use Committees before study initiation (ACUC; Protocol VB043). Animals were closely monitored daily for any signs of illness, and appropriate medical care was provided as needed. Animals were socially housed per the approved ACUC protocol and social compatibility. All clinical procedures, including biopsy collection, administration of anesthetics and analgesics, and euthanasia, were carried out under the direction of a laboratory animal veterinarian. Steps were taken to ensure the welfare of the animals and minimize discomfort of all animals used in this study. Animals were fed daily with a fresh diet of primate biscuits, fruit, peanuts, and other food items to maintain body weight or normal growth. Animals were monitored for mental health and provided with physical enrichment, including sanitized toys, destructible enrichment (cardboard and other paper products), and audio and visual stimulation.

### Animal inoculation and treatments
Nineteen male and female rhesus macaques uninfected with SIV/SHIV, as demonstrated by several consecutive negative PCRs and seronegative for simian T-cell lymphotropic virus 1 at the initiation of the study, were randomized into three groups based on sex, age, weight, and their prior enrollment in other studies (Supplementary Table 1)[39]. Five animals from the α-CD8/NK/Clodrosome®/HTLV-1A/C$_{oI-L}$ group and ten animals from α-CD8/NK/Clodrosome®/HTLV-1A were treated for three consecutive days (Days −3, −2, −1) with an anti-CD8 monoclonal antibody the clone MT807R1 targeting the α/α chain of the CD8$^+$ lymphocytes and NK cells, in addition to a single dose of Clodrosome® delivered in Liposome (Day −1) targeting the phagocytic cells. Four control animals in the IgG/Liposome/HTLV-1A/C$_{oI-L}$ group were treated for three consecutive days (Days −3, −2, −1) with the isotype control antibody IgG OKT3, reactive against the human CD3 molecules, in addition to a single dose (Day −1) of Encapsome® corresponding to an empty Liposome. Both antibodies, anti-CD8 and IgG OKT3, were purchased from the NHP Reagent Resource Program (University of Massachusetts Medical School, Worcester, MA), while Clodrosome® (cat. #CLD-8909) and Encapsome® (cat. #CLD-8910) were purchased from Encapsula NanoSciences (Brentwood, TN). All treatments were administered intravenously at 5 mg/kg/dose/day prior to the intravenous inoculation of $1 \times 10^8$ or $1.5 \times 10^8$ lethally γ-irradiated 729.6 lymphoblastoid B-cell lines producing either HTLV-1A or HTLV-1A/C$_{oI-L}$, respectively. The inoculated cell number was normalized for p19 Gag antigen production and viral DNA level to reflect the amount used in our

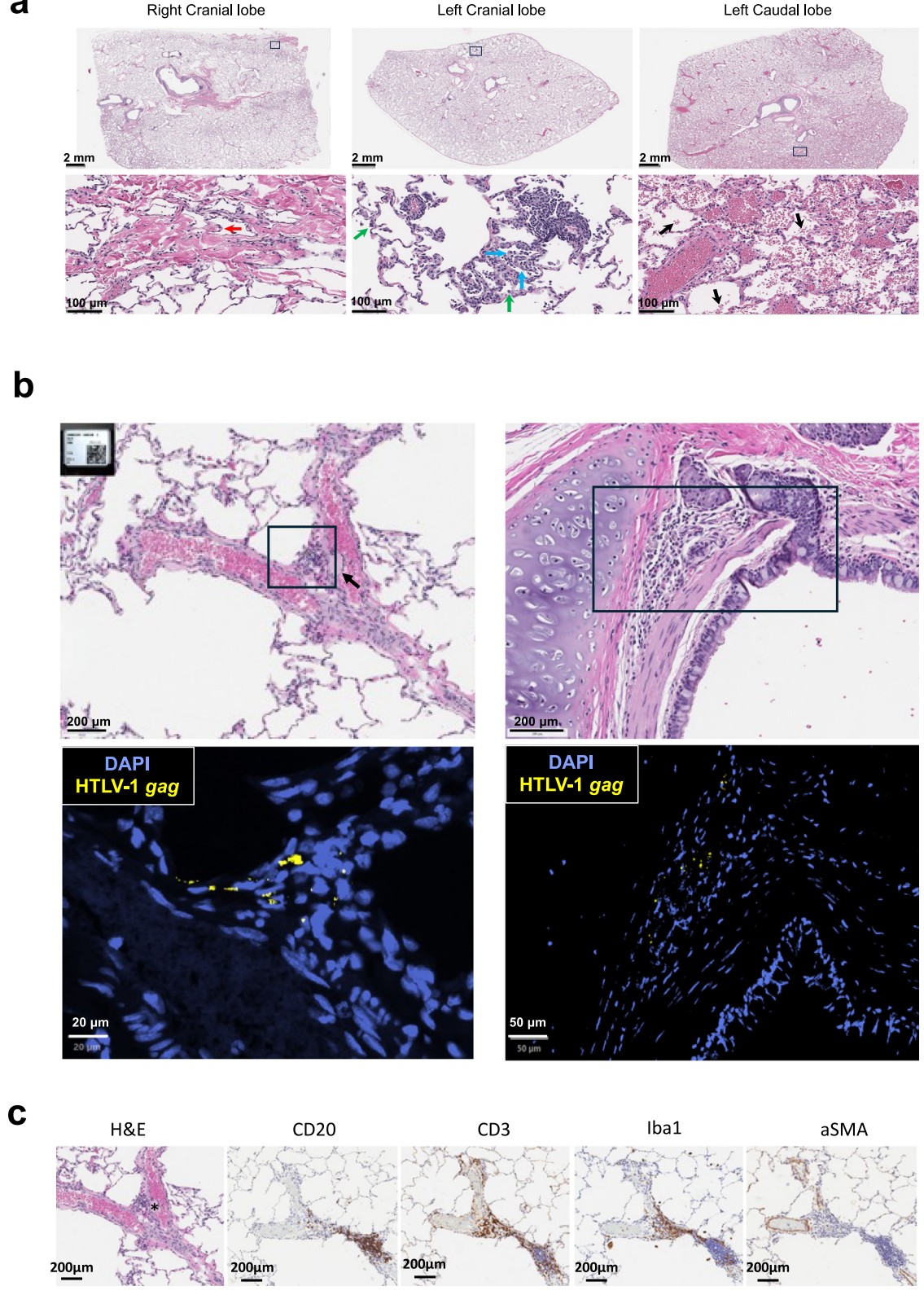

previous studies[33,35,39]. Animals were monitored for over 21 weeks post viral inoculation and then euthanized to study viral dissemination and pathogenesis in tissues except for TMN, DG8Z, TiT, RA6, TRE, TZW, RH5, RKF, and DHF6 that were monitored for over 48 weeks. Except for randomization purposes, the sex of the animals was not considered as discriminating factor in our study. Since the aim of our study was to investigate the infectivity and the pathogenicity of the newly constructed chimeric molecular clone, both male and female mice were used in each group.

**Biosafety statement**

All experiments involving HTLV-1A or HTLV-1A/C$_{ol-L}$ were conducted in a Biosafety Level 2 (BSL-2) laboratory with BSL-3 practices in accordance with the institutional biosafety guidelines. Standard

**Fig. 5 | Lung Histopathology in HTLV-1A/C$_{ol\text{-}L}$ infected animals. a** H&E-stained lung lobe biopsies from three HTLV-1A/C$_{ol\text{-}L}$ infected animals (TMN, DG8Z, and TiT). Histopathology images showing peripheral fibrosis (black box, left upper panel) characterized by increased collagenous stroma, expanded alveolar septa (red arrows, left lower panel) in the right cranial lobe of TMN. The middle panels show interstitial pneumonia, characterized by focal thickening of the alveolar septa with increased immune infiltrates in the left cranial lobe of DG8Z (black box, upper middle panel) with lymphocytes and macrophages within alveolar spaces (green arrows, low middle panel) and the rare type 2 pneumocyte hyperplasia (blue arrows, lower middle panel). The right panels depict the pulmonary hemorrhage within the alveolar spaces in the left caudal lobe of TiT (black arrows, lower right panel). Scale bars 2 mm and 100 µm for low (upper panels) and high (lower panels) magnification, respectively. **b** Representative low and high-magnification *gag* RNAscope images (upper and lower panels, respectively) from the right cranial lobe of animal TMN (left panels) and the left caudal lobe of TiT (right lobes) stained with HTLV-1 Gag probe (yellow) alongside DAPI (blue) for nuclear counterstain. Low and high-magnification scale bars 300 µm and 20 or 50 µm, respectively. **c** Images from TMN right cranial lobe showing single-antibody immunohistochemistry for CD20 (B-cell marker), CD3 (T-cell marker), Iba1 (macrophage/monocyte marker), and SMA (activated fibrogenic cell marker). The H&E stain was performed on the same section. Asterisk (*) designates a vessel containing a fibrin thrombus with re-endothelialization. Scale bars 300 µm. Immunohistochemistry and RNAscope assays were used to quantify and characterize immune cells in whole lung sections for 20 slides from 4 groups (5 slides per group, different lung lobes). Source data are provided as a Source data file.

microbiological practices were followed, including the use of appropriate Personal Protective Equipment (lab coats and gloves), performing aerosol-generating procedures in a certified Biological Safety Cabinet, and proper decontamination of waste materials.

## Plasmid construction and verification

**HTLV-1A/C$_{ol\text{-}L}$ chimeric molecular clone.** An artificially synthesized DNA fragment encompassing a portion of HTLV-1 subtype A envelope and the entire 3′end that spanned the *orf-I, II, III, IV*, and the 3′LTR of the HTLV-1C was introduced into a SalI/EcoRI cassette. The molecular clone HTLV-1A (pAB_D26)[34] was cleaved at the SalI/EcoRI sites and was used as backbone for the construction of the chimeric molecular clone HTLV-1A/C$_{ol\text{-}L}$. A molar ratio of 1:7 (Vector: Insert) was used for the ligation in the presence of Takara Ligation mix (cat. #6023 Takara Bio, San Jose, CA). After 10 min of incubation at room temperature (RT), 3 µl of the ligation mix was used to transform 50 µl of the One Shot Stbl3 Chemically Competent E. coli (cat. #C737303, Thermo Fisher Scientific, Waltham, MA). Briefly, the cells/ligation mix was incubated on ice for 30 min, and the cells were then heat-shocked for 30 s at 42 °C in a water bath. The vial was returned and incubated on ice for another 2 min, and 250 µl of prewarmed SOC medium was then added to the cells and incubated in a shaking incubator for 2 h at 27 °C. The mix was then streaked on a prewarmed kanamycin-resistant plate and incubated at RT for 2 to 3 overnight. The smallest colonies were separately collected, and mini cultures were started by adding 4 mL of Luria Bertani (LB) Broth (cat. #BLF-7030, KD Medical, Columbia, MD) supplemented with 50 µg/mL kanamycin. The cultures were incubated in a shaker at 27 °C for 24 to 48 h. Plasmids were isolated using the Wizard Plus SV Minipreps DNA purification System (cat. #A1460, Promega, Madison, WI) following the manufacturer's instructions. A restriction digestion was then performed using SpeI and NheI/KpnI restriction enzymes. The molecular clones displaying the expected restriction digestion pattern were amplified by Maxiprep using Qiagen Plasmid Maxi Kits (cat. #12162, Qiagen, Germantown, MD) following manufacturer's instructions. The viral genomic sequences were verified by sequencing. The sequence of the pAB_HTLV-1A/C$_{ol\text{-}L}$ chimeric molecular clone was submitted to NCBI GenBank nucleotide database (accession number PP860917). The HTLV-1C viral nucleotide sequence used in the cloning was derived from virus isolated from PBMCs of an infected donor. The nucleotide sequence was obtained from the Laboratory of Dr. Damian JF Purcell[18] (accession nos. PP596271, PP596272, PP596273, PP596274, PP596275, PP596276, PP596277, PP596278, PP596279, PP596280, PP596281, PP596282, PP596283, PP596284, PP596285, PP596286, PP596287, PP596288, PP596289, PP596290, PP596291, PP596292 for all patient proviruses).

**HTLV-1A/C$_{E\text{-}L}$ chimeric molecular clone.** An artificially synthesized DNA fragment encompassing a portion of the polymerase, the entire envelope, and *orf-I* of HTLV-1 subtype C was introduced into a BbvCI/XbaI cassette. The chimeric molecular clone HTLV-1A/C$_{ol\text{-}L}$ described above was cleaved at the BbvCI/XbaI sites and was used as backbone for the construction of the chimeric molecular clone HTLV-1A/C$_{E\text{-}L}$.

**HTLV-1C-LTR-Luc.** An artificially synthesized DNA fragment of the 5′ long terminal repeat (LTR) region of HTLV-1C virus spanning the U3-R-U5 regions was introduced into an XhoI/HindIII cassette. The cassette was then swiped into the LTR-HTLV-1A-Luc at the XhoI/HindIII restriction sites[59]. The bacterial transformation was performed as described above with few changes: i) a molar ratio of 1:3 (Vector:Insert), ii) incubation performed at 37 °C, and iii) ampicillin-resistant plate/media. The isolated plasmids were first verified by restriction digestion using XhoI/HindIII, and the plasmid displaying the expected restriction digestion pattern were sequenced and amplified by Maxiprep. The 5′LTR-HTLV-1C nucleotide sequence used in the cloning was derived from virus isolated from PBMCs of the same infected donor described above.

**pRL-TK-Luc, HTLV-1A-LTR-Luc, and NF-κB-Luc[60] pSDM-12-HA, pSDM-p16A-HA, and pSDM-p16C-HA.** Cassettes were digested with PmeI and BamH1 restriction enzymes and re-cloned into the PmeI and BamH1 digested pSDM101 vector[61] (hereafter pSDM). All constructs were verified by sequencing.

## Cell lines and primary cell culture

**Adherent cell lines.** HEK293T and HeLa cell lines were cultured in Dulbecco's Modified Eagle's Medium (DMEM) (Invitrogen, Carlsbad, CA) supplemented with 1% penicillin/streptomycin and 10% fetal bovine serum (FBS) (cat. #16140071, Life Technology, Carlsbad, CA). Suspension cell lines: Jurkat T-lymphoblastic and monocytic THP-1 cell lines were cultured in a Roswell Park Memorial Institute (RPMI 1640) medium (Invitrogen, Carlsbad, CA) supplemented with 1% penicillin/streptomycin and 10% FBS. Peripheral blood mononuclear cells: PBMCs were separated from blood of healthy human donors, naïve rhesus macaques, or heparinized human neonatal umbilical cord blood by density gradient centrifugation by Ficoll Plaque (GE Healthcare).

**CD4$^+$ cell isolation.** CD4$^+$ cells were isolated from cryopreserved human and non-human primate PBMCs using PE-conjugated antibodies and Anti-PE Microbeads (cat. #130-048-801, Miltenyi Biotec). Briefly, $30 \times 10^6$ PBMCs were thawed, then incubated at 4 °C for 15 min with 155 µl of MACS buffer/tube and 45 µl of PE anti-CD4 (clone L200; cat. #550630, BD Biosciences) antibody. Following incubation, cells were washed with 3 ml buffer per tube. Cells in both tubes were then incubated at 4 °C for 15 min with 240 µl of MACS buffer and 60 µl of anti-PE microbeads for each isolation. Cells were then washed again with 3 ml buffer, resuspended with 500 µl of MACS buffer, and isolated using the AutoMACSpro (Miltenyi Biotec). CD4$^+$ cell positive selection was performed following the Possel program. Mononuclear cells were cultured in RPMI supplemented with 1% penicillin/streptomycin, 20% FBS, and 100 U/ml of purified interleukin-2 (IL-2) (cat. #I2644, Millipore Sigma, Rockville, MD) in a 6-well plate at a density of 1 to $5 \times 10^6$ cells

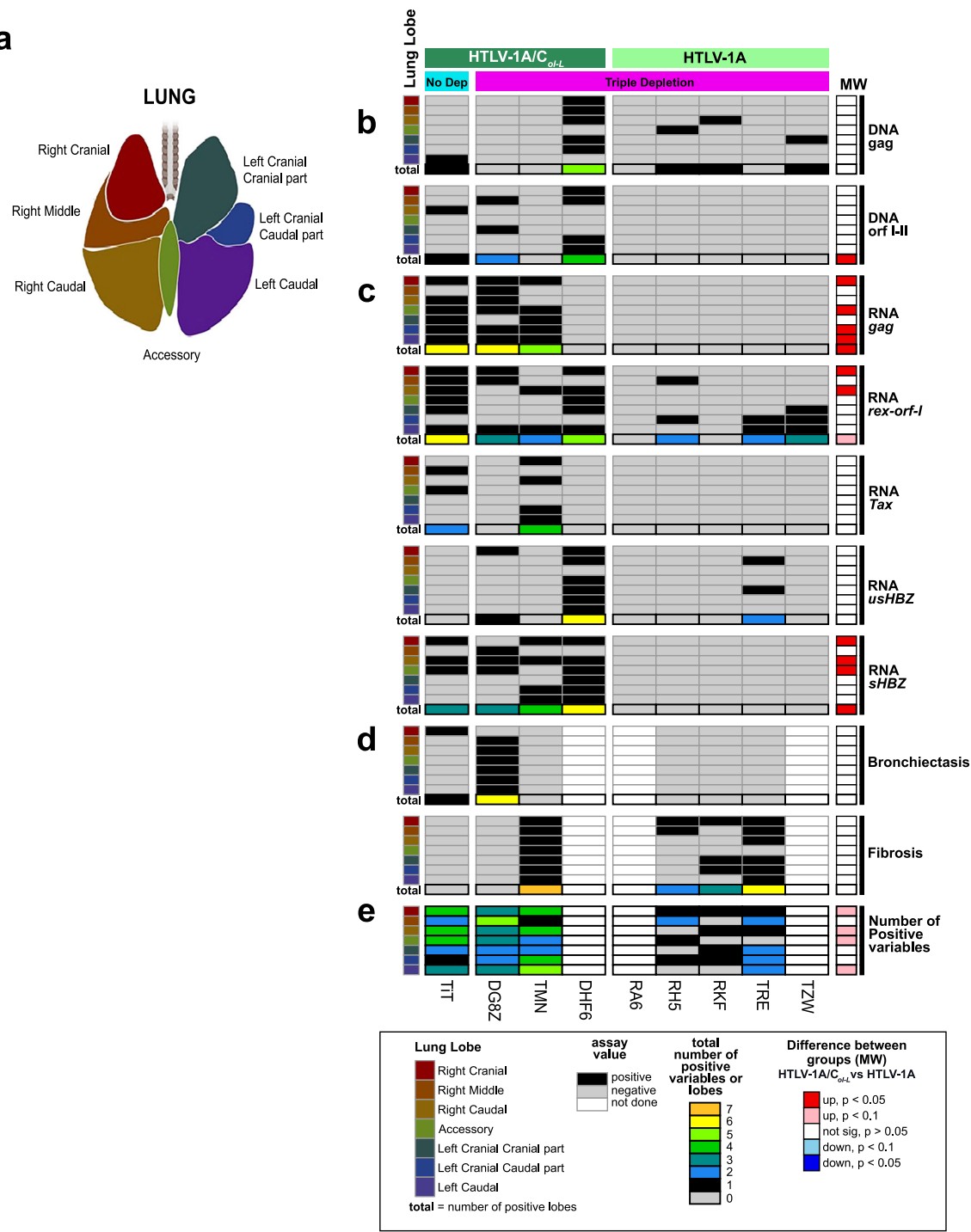

**Fig. 6 | Virus score in the lung of triple-depleted animals exposed to HTLV-1A or HTLV-1A/C_{ol-L}. a** Rhesus macaque lung schematic showing the seven lung lobes (right cranial, right middle, right caudal, accessory, left cranial/cranial part, left cranial/caudal part, and left caudal). **b–e** Summary of the lung virus variables in addition to the histopathology assay. The bronchiectasis and fibrosis data were converted to binary presence/absence to match all DNA/RNA assays. Heatmaps are shown with animals in columns and variables in rows. In addition to the 7 assays and 2 pathology assays, a total number of positive assays for each lung region (bottom row of each block), and the total number of positive lung regions for each assay (**e**) were calculated. The statistical analysis was Mann–Whitney, and p value ranges are reported in the rightmost row-annotation column. Rows are sorted first by Assay, then by lobe. The lung schematic was created in BioRender. Sarkis, S. (2025) https://BioRender.com/qzewdz4. Source data are provided as a Source data file.

per well. The medium was renewed twice a week, and fresh IL-2 was added to the cultures.

## Transfection

HEK293T and HeLa cells were transfected using LipoD293 In vitro DNA transfection Reagent (cat. #SL100668, SignaGen Laboratories, Frederick, MD) following the manufacturer's instructions. Briefly, cells were plated 18 h prior transfection, allowing the cells to reach a 70 to 80% confluency by the time of the transfection. The diluted LipoD293 (LipoD293 + media) was added to the diluted DNA (DNA + media) to a ratio LipoD293 (µl) – DNA (µg) of 3:1. The Mix was briefly vortexed and incubated at RT for 10–15 min. The mix was then added dropwise onto

the media in each well, and the mixture was homogenized by gently swirling the plate. 12 h post-transfection, the LipoD293/DNA complex-containing media was replaced, and the transfection efficacy was checked 24–48 h post-transfection.

## Establishment of stable Jurkat (T-cell) and monocytic THP-1 cell lines expressing viral proteins

Jurkat and THP-1 cell lines were transduced with GFP-expressing lentiviral empty vector control (pSDM) or pSDM expressing p12-HA, p16A-HA, or p16C-HA. Briefly, 80% confluent HEK293FT cells ($6 \times 10^6$ cells) cultured in 10 cm dish, were transfected using LipoD293 transfection reagent as described above with a DNA mix constituted of 12 µg of pSDM, pSDM-p12-HA, pSDM-p16A-HA, or pSDM-p16C-HA in addition to 8 µg of psPAX-2 (Packaging plasmid, Addgene) and 6 µg pMD2.G (Env, Addgene). 48 h post-transfection, supernatants of the different cell cultures were collected, centrifuged for 5 min at $500 \times g$, and filtered through a 0.22 µm filter. Lentiviruses were concentrated by centrifugation at $8000 \times g$ for 3 h at 4 °C. Concentrated lentiviruses were resuspended in 100 µl of RPMI. Jurkat and THP-1 cells were centrifuged for 5 min at $300 \times g$, then resuspended in the virus solution and incubated for 5 min at 37 °C. The mix of cells-lentivirus is then centrifuged for 10 min at $500 \times g$, and the pellet is resuspended in 1 ml of complete medium and transferred in a 12-well plate for 72 h. Transduction efficiency was measured by flow cytometry (FACSCalibur, BD). Additionally, the expression of viral proteins (p12, p16A, and p16C) was verified by Western blot for HA.

## Spin-infection

Heparinized human neonatal umbilical cord blood and CD4$^+$ T-cells were spin-infected with γ-lethally irradiated 729.6 lymphoblastoid B-cell lines producing either HTLV-1A or HTLV-1A/C$_{ol-L}$ viruses. The number of donor cells (729.6 B-cells) used to infect the acceptor cells (primary cells) was normalized for p19 Gag production and viral DNA level, and the ratio of acceptor:donor was 1:0.5 and 1:1 for HTLV-1A and HTLV-1A/C$_{ol-L}$, respectively. Briefly, after isolation, 1 to $5 \times 10^6$ primary cells were washed twice with RPMI supplemented with 10%FBS and 1% penicillin /streptomycin and the pellets were resuspended with the γ-lethally irradiated 729.6 B-cell lines in a final volume of 4 ml of RPMI supplemented with 10%FBS and 1% penicillin /streptomycin in the presence of polybrene with a final concentration of 8 µg/ml (cat. #sc-134220, Santa Cruz Biotechnology, Dallas, Texas). The donor/acceptor-cell mix was then centrifuged at $2000 \times g$ at RT for 1 h, the supernatant was discarded, and the pellet was resuspended in RPMI medium supplemented with 20% FBS, 1% penicillin/streptomycin, and 100 U of IL-2. The media was changed twice a week, and fresh IL-2 was added to the primary culture.

## Establishment of the HTLV-1A/C$_{ol-L}$ producer cell line by electroporation

729.6 lymphoblastoid B-cell lines (hereafter 729.6 B-cells) (kindly provided by Patrick Green, Department of Veterinary Biosciences, Ohio State University), were transfected using the Amaxa Cell Line Nucleofector Kit V (cat. #VCA-1003, Lonza, Morristown, NJ) and following manufacturer's instructions. Briefly, $5 \times 10^6$ of low-passaged 729.6 B-cells were centrifuged at $600 \times g$ for 10 min, and the pellet was resuspended in 100 µl of Nucleofector Solution. The cell/Nucleofector solution mix was then mixed with 5 µg of the pAB HTLV-1A/C$_{ol-L}$ molecular clone and then transferred into a certified cuvette. Electroporation was performed using the pre-set optimized M-013 program on the Amaxa-Nucleofector II (Amaxa Biosystems). After the electroporation, 500 µl of RPMI media complemented with 20% FBS and 1% penicillin/streptomycin is immediately added into the cuvette, and the cells were incubated at RT for 10 min before transferring them into larger flasks containing 4.5 ml of media and incubated at 37 °C. 48 h post-electroporation the cells were collected by centrifugation

and resuspended in RPMI media supplemented with 10%FBS, 1% penicillin/streptomycin, and 100 mg/ml of Geneticin (cat. #ant-gn-5, Invivogen, San Diego, CA), used as a selection agent. Following 4–5 weeks of selection period, viable cells were expanded in culture for further analysis.

## Western blot

Cell lysates were separated by SDS-PAGE (cat. #NP0321, NuPAGE 4–12% Bis-Tris Protein Gels, Thermo Fisher Scientific, Waltham, MA) and transferred to a polyvinylidene difluoride membrane (Immobilon-P PVDF, Millipore Sigma, St. Louis, MO). The membranes were incubated overnight at 4 °C with primary antibodies to HTLV-1 p24 Gag (Mouse, Applied Biological Laboratories cat. #4310, 1:1000), HTLV-1 gp46 envelope (Mouse, Creative Biolabs, cat. #CBMAB-V208-1154-FY, 1:100), Tax (Tab172, 1:100 ref. 34; Tax-LT-4, Mouse, Millipore Sigma cat. #MABF3063, 1:1000; Tax-1A3, Mouse, Abcam cat. #ab26997, 1:1000), GFP (Mouse, Thermo Fisher Scientific, cat. #MA5-15256), β-actin (Rabbit Cell Signaling Technology, D6A8, 1:1000), GAPDH (Rabbit Cell Signaling Technology, D3U4C, 1:1000), HA (Rabbit, Cell Signaling Technology, C29F4, 1:1000) in PBS containing 0.1% Tween 20 and 0.25% milk. Membranes were washed in PBS 0.1% Tween and exposed to a horseradish peroxidase-conjugated goat secondary anti-Mouse or anti-Rabbit antibody (Thermo Fisher Scientific NA931 and NA934, respectively; 1:10.000). Blots were developed with the use of either SuperSignal West Pico Plus or West Femto Maximum sensitivity substrate chemiluminescent detection system (cat. #34579 and 34095, respectively, Thermo Fisher Scientific). Proteins were visualized by chemiluminescence using a ChemiDoc Imaging System (Bio-Rad Laboratories, Hercules, CA). Full scan blots are in the Source data file.

## Dual-Glo luciferase assay

HEK293T cells were transfected with HTLV-1A-LTR-Luc, HTLV-1C-LTR-Luc, or NF-κB-Luc reporter plasmids and either HTLV-1A or HTLV-1A/C$_{ol-L}$ molecular clones with a reporter-molecular clone ratio of 2:1 and a LipoD293-DNA ratio of 3:1. The amount of DNA transfected was equalized by addition of a control vector (pcDNA 3.1). All the transfections were carried out in the presence of a pRL-TK-Luc vector in order to normalize the results for the transfection efficiencies. 48 h post-transfection, cells were extracted with Passive Lysis Buffer (Promega, Milwaukee, WI) and reporter activities were assayed using Dual-Glo reagent (cat. #E2940, Promega, Milwaukee, WI) following manufacturer's instructions. Briefly, to each well containing 5 µl of cell lysate, 25 µl of Dual-Glo Reagent is added, and the mix is incubated on a shaker at RT for at 10 min then the firefly luminescence is measured in a luminometer. A volume of Dual-Glo Stop & Glo Reagent equal to the volume of Dual-Glo Reagent is then added to each well, and the mix is incubated at RT on a shaker for 10 min, and the *Renilla* luminescence is measured. Luciferase assays were performed with the VictorX4 2030 multilabel plate reader (PerkinElmer, Waltham, MA). Results were normalized for Renilla luciferase activity, and fold change induction was calculated by dividing each luciferase activity value by values of the cells transfected in the absence of the molecular clone plasmids (pcDNA 3.1).

## RNA extraction and reverse transcription PCR

Total RNA was extracted from the stably infected cell lines, stably transduced Jurkat and THP-1 cell lines, as well as from macaque lung lobes using RNeasy plus mini kit (cat. #74034, Qiagen, Germantown, MD) following the manufacturer's instructions. Briefly, the cells were harvested and lysed in the RLT Plus buffer. The homogenized lysates are then transferred to a gDNA eliminator spin column and centrifuged for 30 s at $8500 \times g$. 70% ethanol is then added to the flow-through, and the mix is transferred to the RNeasy spin column and centrifuged. The column is then washed 3 consecutive times using once the RW1 and twice the RPE buffers. Following the last wash, the column is air-dried

by centrifugation at high speed for 1 min. The RNAs are then eluted in 50 µl of nuclease-free water by centrifugation. The eluted ARNs are quantified using a Nanodrop ND-2000 apparatus (Thermo Fisher Scientific). RNA samples were then subjected to Reverse transcription using QuantiTect Reverse transcription Kit according to the manufacturer's instructions (cat. #205311; Qiagen, Germantown, MD). Briefly, 500 ng of total RNA was mixed with gDNA wipeout Buffer and RNase-free water. The mix is incubated for 3 min at 42 °C, then placed immediately on ice. A second mix of reverse transcriptase, reverse transcriptase buffer, and reverse transcriptase primers is then added to the first mix and incubated for 30 min at 42 °C, followed by an incubation at 95 °C for 3 min to inactivate the reverse transcriptase. The cDNA materials are then stored at −80 °C until use. Prior to RNA extraction from lung lobes, the tissues preserved in RNAlater™ were thawed on ice and centrifuged at $800 \times g$ for 3 min at 4 °C. The supernatant was discarded and 600 µl of RPE buffer supplemented with β-Mercaptoethonal (10 µl in 1 ml) was added to each sample. The tissues were resuspended by pipetting until dissolution, and the lysates were then centrifuged through a QIAshredder (cat. #79654; Qiagen, Germantown, MD) homogenizer spin column at full speed for 2 min.

### DNA extraction and viral DNA detection

Genomic DNA from the PBMC, bone marrow, and biopsies from the Lymphnodes the BAL, and the lung lobes were isolated from all animals during the time course of the study except the lungs lobes collected at the time of the sacrifice, using the DNeasy Blood and Tissue Kit (Qiagen) following the manufacturer's instructions. 100 ng of DNA were used as templates for the first round of PCR amplification. Three microliters of the PCR reaction were then used as a template for nested PCR. The PCR conditions used were 94 °C for 2 min, followed by 35 cycles of 94 °C for 30 s, 55 °C for 30 s, 68 °C for 60 s, and a final extension at 68 °C for 7 min, and a hold at 4 °C. Platinum High Fidelity PCR SuperMix (Invitrogen, Carlsbad, CA) was used according to the manufacturer's protocol. Correctly sized amplicons were identified by 1% agarose gel electrophoresis.

### Primers

All primers used in the study (Supplementary Table 5) were designed using Primer3 software (v. 0.4.0) and produced by Eurofins Genomics LLC (Louisville, Kentucky).

### Transcriptional characterization of HTLV-1A/C$_{ol-L}$

The primers (Supplementary Table 5) used to detect the different mRNA viral species in the 729.6 B-cells stably infected cell lines were designed using Primers3 (v.0.4.0); either on the exon sequence for *gag-pro-pol, tax-orf-II, usHBZ*, and *β-actin* or boundary spanning primers overlapping the splice sites of the singly (*env, orf-I, orf-II, p21$^{rex}$*, and *sHBZ*) and doubly spliced (*tax/rex* and *rex-orf-I*) mRNAs. The PCR was performed in the presence (+) or absence (−) of the Reverse Transcriptase (see section above). The PCR conditions used for amplification were 94 °C for 2 min, followed by 35 cycles of 94 °C for 30 s, 57 °C for 30 s, 68 °C for 60 s, and a final extension at 68 °C for 7 min, and a hold at 4 °C. Platinum High Fidelity PCR SuperMix (cat. #12532016, Invitrogen, Carlsbad, CA) was used according to the manufacturer's protocol. The PCR products were separated on a 1% agarose gel and visualized by ethidium bromide staining. Correctly sized amplicons were identified by 1% agarose gel electrophoresis. Sanger sequencing was carried at Center for Cancer Research Genomics Core at the National Cancer Institute, NIH; on all transcripts to check for any depletion or mutation that would affect the amino acid sequence of the encoded protein. The failure to detect a signal in the absence of reverse transcriptase confirmed that the mRNA is free of plasmid or genomic DNA contamination. Full scan gels are in the Source data file.

### HTLV p19 antigen ELISA

p19 Gag detection was measured by ELISA assay in the culture supernatants of $1 \times 10^6$ cells washed and seeded for 24 h in a 24-well plate in 1 mL of complete RPMI following the manufacturer's instructions (ZeptoMetrix, Buffalo, NY). The p19 Gag detection in infected primary cell cultures was measured as accumulation of p19 Gag antigen without normalization of the cell number.

### HTLV-1 p24 antibody titer

HTLV-1 p24 antibodies in plasma samples from macaques were detected and quantified against purified HTLV-1 p24 protein using an ELISA assay. Each well of the ELISA plate was coated with HTLV-1 p24 core antigen (Prospec, cat. #hiv-108-c) at 500 ng per well and incubated overnight at 4 °C. Wells were then emptied and blocked with Superblock Blocking Buffer in PBS (cat. #37515, Thermo Fisher Scientific, Waltham, MA) for 1 h at room temperature. Serially diluted samples were added to the wells and incubated overnight at 4 °C. The plate was then washed with PBS Tween 20 and incubated for 1 h at 37 °C with the diluted goat anti-human IgG HRP (Kirkegaard & Perry Lab Inc., Gaithersburg, MD). The plate was washed, and the 1-Step Ultra TMB-ELISA (cat. #34029, Thermo Fisher Scientific, Waltham, MA) was added to the wells and incubated for 30 min at room temperature. The reaction was stopped by adding $_{KPL}$ TMB Stop Solution (cat. #5150-0019), and the plate was read at 450 nm using VictorX4 2030 multi-label plate reader (PerkinElmer, Waltham, MA).

### HTLV serology

Reactivity to specific HTLV-1 viral antigens in the plasma of the animals was detected with the use of a commercial HTLV-1 western immuno-blot assay (GeneLabs Diagnostics, Redwood City, CA) following the manufacturer's instructions. Briefly, pre-hydrated strips are incubated for 1 h at RT on a rocking platform with 20 µL of animal sera in the presence of a blotting buffer. The strips are washed three times, then incubated for 1 h with the working conjugated solution. After the washes the Substrate solution is added to the strips and incubated at RT for 15 min. The strips are then washed and dried.

### Proximity extension assay (PEA)

**Plasma and BAL samples.** Protein quantification was executed employing the Olink® Target 48 Cytokine panel* (Olink Proteomics AB, Uppsala, Sweden) in accordance with the manufacturer's protocols. This method leverages the Proximity Extension Assay (PEA) technology, as extensively detailed by Assarsson et al.[62]. This specific PEA methodology enables the concurrent assessment of 45 distinct analytes. Briefly, we used pairs of oligonucleotide-labeled antibody probes, each tailored to selectively bind to their designated protein targets. Probe pairs mix was incubated with 1 µl of plasma or BAL fluid. Probes that encountered their cognate proteins are then in close spatial proximity, and their respective oligonucleotides engage in pairwise hybridization. A DNA polymerase was used to amplify the polymerized DNA and to create distinct PCR target sequences. Subsequently, we detected and quantified these newly formed DNA sequences through utilization of a microfluidic real-time PCR platform, specifically the Biomark HD system by Fluidigm (Olink Signature Q100 instrument). Data validation to uphold data integrity was conducted with the Olink NPX Signature software, specifically designed for the Olink® analysis: the application was used to import data from the Olink Signature Q100 instrument and process the data. Data normalization procedures were executed employing an internal extension control and calibrators, thereby effectively mitigating any inherent intra-run variability. The ultimate assay output was reported in pg/ml, predicated upon a robust 4-parameter logistic (4-Pl) fit model, thereby ensuring precise absolute quantification. Comprehensive insights into the assay's validation parameters, encompassing limits of detection,

intra- and inter-assay precision data, and related metrics, are available at www.olink.com.

Output from the Olink software was further processed to extrapolate values for samples that were below the lower limit of quantification (LLOQ) and above the upper limit of quantification (annotated as >ULOQ). The Olink software interpolates values below the LLOQ through fitting the 4-Pl model to a distinct minimum limit of detection for each plate of samples run, and values below this interpolation range are set to NaN. Since these values are below the limit of detection but not truly missing, for each assay, we determined a universal below detection value by taking the mode LLOQ across 22 plates, divided by 10,000 (which was below all interpolated values in this extensive historical dataset), and set all NaN to this assay-specific below detection value. Samples that were above the ULOQ were set to the ULOQ value for the indicated target assay from the plate on which the sample was run. Finally, true missing values annotated as No Data were converted to NA to be systematically treated as missing.

**THP-1 stably transduced cells.** THP-1 p12-HA-GFP or THP-1 p16C-HA-GFP stable cell lines were plated in 24-well plate at a density of $1 \times 10^6$ cells/mL. 24 h post-treatment with PMA, the culture supernatant was removed, and a fresh medium was added. 24 h later, the culture supernatant was collected and stored at −80 °C.

**Flow cytometry analysis**
For the whole blood cell phenotyping, 100 µl of fresh EDTA whole blood was stained with Fluorochrome-conjugated mAbs. 1µl of the following antibodies were used: FITC anti-CD8 (clone DK25; cat. #FCMAB176F; EMB Millipore Corp.), BB700 anti-CD14 (clone M5E2; cat. #745790; BD Biosciences), PE-Cy5 anti-CD95 (clone DX2; cat. #305610; BioLegend), PE-Cy7 anti CD159 (NKG2a) (clone Z199; cat. #B10246; Beckman Coulter), APC anti-CD66abce (clone TET2; cat. #130-118-539; Miltenyi Biotec), Alexa 700 anti-CD3 (clone SP34-2; cat#.557917; BD Biosciences), APC-Cy7 anti-CD11b (clone ICRF44; cat. #557754; BD Biosciences), BV421 anti-CD16 (clone 3G8; cat. #562874; BD Biosciences), BV570 anti-CD20 (clone 2H7; cat. #302332; BioLegend), BV750 anti-CD4 (clone L200; cat. #747202; BD Biosciences), BV786 anti-CD45 (clone D058-1283; cat. #563861; BD Biosciences), BUV496 anti-CD28 (clone CD28.2; cat. #741168; BD Biosciences), BUV563 anti-CD49d (clone 9F10; cat. #749455; BD Biosciences), BUV661 anti-HLA-DR (clone G-46-6; cat. #612980; BD Biosciences), BV711 anti-CD11c (clone B-ly6; cat. #741139; BD Biosciences), BV650 anti-CD123 (clone 7G3; cat. #572392; BD Biosciences), BUV805 anti-CD8 (clone SK1; cat. #612889; BD Biosciences). Blue LIVE/Dead viability dye (cat. #L23105; Thermo Fisher Scientific, Waltham, MA) was used to exclude dead cells. T-cell lineages were identified following the gating strategy; i) Singlets/Live/CD45$^+$/CD20$^-$CD14$^-$/CD3$^+$CD4$^+$ for CD4$^+$ T-cells, ii) CD4$^+$CD8$^-$/CD95$^+$ for CD4$^+$ memory helper T-cells, iii) CD4$^+$CD95$^-$ for CD4$^+$ naïve helper T-cells, iv) CD4$^+$CD95$^-$CD28$^-$ for CD4$^+$ effector memory helper T-cells, v) CD4$^+$CD95$^-$CD28$^+$ for CD4$^+$ central memory helper T-cells, vi) CD3$^+$CD4$^-$CD8$^+$ for CD8$^+$ cytotoxic T-cells (CTLs), vii) CD3$^+$CD8$^+$CD95$^+$ for memory CTLs, viii) CD8$^+$CD95$^-$ for naïve CTLs, ix) CD8$^+$CD95$^-$CD28$^-$ for effector memory CTLs, x) CD8$^+$CD95$^-$CD28$^+$ for central memory CTLs. NK and neutrophils were identified following the gating strategies Singlets/Live/CD45$^+$/CD3$^-$CD20$^-$/CD14$^-$/CD8$^+$NKG2a$^+$, and Singlets/Live/CD45$^+$/CD20$^-$CD3$^-$/CD8$^-$/CD123$^-$CD11c$^-$/CD14$^-$CD16$^-$/CD66abce$^+$, respectively. Monocyte populations were identified as Singlets/Live/CD45$^+$/HLA-DR$^+$CD20$^-$/CD3$^-$CD8$^-$ and differentiated by the expression of CD14 and CD16[63]. Classical monocytes were identified as CD14$^+$CD16$^-$, intermediate monocytes as CD14$^+$CD16$^+$, and non-classical monocytes as CD14$^-$CD16$^+$. Briefly, following the 30 min staining at RT, the red blood cells were lysed by incubating the samples with the BD FACS Lysing solution (cat. #349202 BD Biosciences, San Jose, CA) for 10 min at RT. Samples were washed with PBS and resuspended in 1% ultrapure

formaldehyde (cat. #1008B-10 Tousimis, Rockville, MD). Flow cytometry acquisitions were performed on a FACSymphony A5 and examined using FACSDiva software (BD Biosciences) by acquiring all stained cells. Data was further analyzed using FlowJo v10.1 (TreeStar, Inc., Ashland, OR).

To measure neutrophils, monocytes, myeloid dendritic cells (mDC), and plasmacytoid dendritic cells (pDC) in bronchoalveolar lavage (BAL) and whole blood of the animals, 200 µL of whole EDTA blood and $1 \times 10^6$ cells freshly isolated from BAL were stained with Blue LIVE/DEAD viability dye (cat. #L34962, Thermo Fisher Scientific) to exclude dead cells. 5µl The following antibodies were used for cell surface staining: FITC anti-CD66abce (clone TET2; cat. #130-116-522; Miltenyi Biotec), BB700 anti-CD162 (clone KPL-1; cat. #745768; BD Biosciences), Alexa 700 anti-CD3 (clone SP34-2; cat. #557917; BD Biosciences), Alexa 700 anti-CD20 (clone 2H7; cat. #560631; BD Biosciences), APC-Cy7 anti-CD11b (clone ICRF44; cat. #47-0118-42; Invitrogen™), BV480 anti-CD11c (clone 3.9; cat. #748269; BD Biosciences), BV650 anti-CD8 (clone RPA-T8; cat. #563821; BD Biosciences), BV750 anti-CD206 (clone 19.2; cat. #746891; BD Biosciences), BV786 anti-CD45 (clone D058-1283; cat. #563861; BD Biosciences), BUV395 anti-123 (clone 7G3; cat. #564195; BD Biosciences), BUV496 anti-CD16 (clone 3G8; cat. #612944; BD Biosciences), BUV563 anti-CD163 (clone GH1/61; cat. #741402; BD Biosciences), BUV661 anti-HLA-DR (clone G-46-6; cat. #612980; BD Biosciences), BUV737 anti-CD64 (clone 10.1; cat. #564426; BD Biosciences), BUV805 anti-CD14 (clone M5E2; cat. #612902; BD Biosciences). Subsequently, cells were permeabilized with Foxp3/Transcription Factor Staining Buffer Set (Invitrogen, cat. #00-5523-00) according to manufacturer's recommendation. The following antibodies were used for intracellular staining: PE anti-MPO (clone MPO455-8E6; cat. #12-1299-42; Invitrogen™), BV421 anti-IL-8 (clone G265-8; cat. #563310; BD Biosciences), BV605 anti-TNF-α (clone mAB11; cat. #502936; BioLegend), and BV711 anti-IL-10 (clone JES3-9D7; cat. #564050; BD Biosciences). Samples were washed with PBS and resuspended in 1% ultrapure formalin (cat. #1008B-10 Tousimis, Rockville, MD). Flow cytometry acquisitions were performed on a FACSymphony A5 and examined using FACSDiva software (BD Biosciences) by acquiring all stained cells. Data was further analyzed using FlowJo v10.1 (TreeStar, Inc., Ashland, OR). In the blood, (i) neutrophils were identified following the gating strategies Singlets/Live/CD45$^+$/CD3$^-$CD20$^-$CD8$^-$/CD123$^-$CD11c$^-$/CD14$^-$CD16$^-$/CD66abce$^+$ cells[64], (ii) monocytes were identified as Singlets/Live/CD45$^+$/CD3$^-$CD20$^-$CD8$^-$/HLA-DR$^+$/FSC-A$^{low}$SSC-A$^{low}$ and differentiated by the expression of CD14 and CD16 as previously published[65], classical monocytes (CD14$^+$CD16$^-$), Intermediate monocytes (CD14$^+$CD16$^+$), and non-classical as (CD14$^-$CD16$^+$); (iii) pDC and mDc were identified following the gating strategies Singlets/Live/CD45$^+$/CD3$^-$CD20$^-$CD8$^-$/HLA-DR$^+$/CD14$^-$/CD123$^+$CD11c$^-$ cells and Singlets/Live/CD45$^+$/CD3$^-$CD20$^-$CD8$^-$/HLA-DR$^+$/CD14$^-$/CD123$^-$CD11c$^+$ cells, respectively[66]. In the BAL, (i) neutrophils were identified following the gating strategies Singlets/Live/CD45$^+$/CD206$^-$/CD163$^-$/CD3$^-$CD20$^-$CD8$^-$/CD123$^-$CD11c$^-$/CD14$^-$CD16$^-$/CD66abce$^+$ cells, (ii) infiltrated monocytes were identified as Singlets/Live/CD45$^+$/CD206$^-$/CD163$^-$/CD3$^-$CD20$^-$CD8$^-$/HLA-DR$^+$/FSC-A$^{low}$SSC-A$^{low}$ and differentiated by the expression of CD14 and CD16 as previously published, classical monocytes (CD14$^+$CD16$^-$), Intermediate monocytes (CD14$^+$CD16$^+$), and non-classical as (CD14$^-$CD16$^+$); (iii) pDC and mDc were identified following the gating strategies Singlets/Live/CD45$^+$/CD206$^-$/CD163$^-$/CD3$^-$CD20$^-$CD8$^-$/HLA-DR$^+$/CD14$^-$/CD123$^+$CD11c$^-$ cells and Singlets/Live/CD45$^+$/CD3$^-$CD20$^-$CD8$^-$/HLA-DR$^+$/CD14$^-$/CD123$^-$CD11c$^+$ cells, respectively.

**Efferocytosis assay**
The effect of the expression of p12 and p16C in apoptotic cells and how it affects the frequency of cells conducting efferocytosis was assessed by Efferocytosis Assay kit (cat. #601770, Cayman Chemical Company, Ann Arbor, MI) and CellTrace™ Far Red (cat. #C34564, Thermo Fisher

Scientific, Waltham, MA), following manufacturers' instructions. The efferocytosis assay was conducted using either monocytic THP-1 cell line or primary CD14⁺ cells isolated from human PBMCs as effector efferocytes, and stably transduced Jurkat cell lines expressing either GFP or p16C-HA-GFP or p12-HA-GFP as target cells.

**Effector cells.** CD14⁺ monocytes were isolated from cryopreserved healthy human donors' PBMCs (n = 3) using human CD14 MicroBeads (cat. #130-050-201, Miltenyi Biotec) and following manufacturer instructions. Briefly, $30 \times 10^6$ PBMCs were thawed and incubated with 60 μl microbeads and 240 μl buffer at 4 °C for 15 min. Cells were then washed with 3 ml buffer and resuspended in 500 μl of buffer. Positive selection was performed using the AutoMACSpro (Miltenyi Biotec) following the Possel program. At the end of the separation, CD14⁺ cells were resuspended in R10 (RPMI media, supplemented with 10% FBS and 1% antibiotic/antimycotic), counted, and washed once with PBS.

Monocytic THP-1 cells grown in R10 were counted and washed once with PBS. Washed THP-1 and CD14⁺ cells were stained with CytoTell™ Blue provided in the Efferocytosis Assay kit by following manufacturer's instructions. Briefly, cells were resuspended in buffer $(1 \times 10^7 \text{ cells/ml})$, an equal volume of buffer containing 2X CytoTell™ Blue (stock diluted 1:200 in kit buffer) was added, incubated at 37 °C for 30 min, washed three times with R10, resuspended in R10, and used for the efferocytosis assay.

**Target cells.** Stably transduced Jurkat-GFP and Jurkat-p16C-HA, and Jurkat-p12-HA express GFP; therefore, the dye CFSE contained in the Efferocytosis Assay kit could not be used. Target cells were stained with CellTrace™ Far Red following manufacturer instructions (cat. #C34564, Thermo Fisher Scientific, Waltham, MA).

Briefly, $40 \times 10^6$ cells per cell line were washed twice with PBS, stained with Far Red staining solution (diluted 1:1000 with PBS) for 20 min at 37 °C. Cells were then washed twice with R10 and treated with apoptosis inducer. The apoptosis of the target cell lines was induced by treatment with Staurosporine Apoptosis inducer provided in the Efferocytosis Assay kit. Briefly, cells were resuspended in R10 media containing Staurosporine (stock diluted 1:1000) and incubated at 37 °C for 3 h. At the end of incubation, cells were washed twice with R10, resuspended $1 \times 10^6$ cells/ml in R10, and used for the efferocytosis assay.

**Coculture of target and effector cells.** THP-1 or CD14⁺ effector cells and Jurkat apoptotic target cells were cultured alone (as controls) or cocultured at a ratio of one effector cell to three target apoptotic cells. For each THP-1 and human donor the experiment was done coculturing the cells in triplicate, with each tube containing 200,000 effector cells and 600,000 target cells. Cells were incubated at 37 °C for 2, 4, 8, 12, 18, and 24 h. At the end of the coculture, cells were washed with PBS, fixed with 1% paraformaldehyde in PBS, and acquired with a flow cytometer. Flow cytometry acquisitions were performed on a FAC-Symphony A5 and examined using FACSDiva software (BD Biosciences) by acquiring all stained cells. Data were further analyzed using FlowJo v10.1 (TreeStar, Inc.). The frequency of CD14⁺ efferocytes was determined as the frequency of GFP⁺ cells (target cells) in the Cyto-Tell™ Blue⁺ cells (effector cells). The CellTrace™ Far Red was not used in the gating strategy to focus on the GFP⁺ cells that express the HTLV protein. Therefore, representing the frequency of effector cells that engulfed the apoptotic target cells. Gating strategy: FSC/SSC/Single cells/CytoTell™ Blue⁺/GFP⁺.

## Immunofluorescence
HeLa cells transfected with PSDM p12 or p16C were cultivated and stained in μ-Slide 8 Well (ibidi) plate. Cells were fixed with freshly prepared 4.0% paraformaldehyde in PBS at RT for 5 min and rinsed thrice with PBS. Cells were permeabilized with PBS containing 0.5%

Triton X-100 for 5 min at room temperature, rinsed thrice with PBS, and blocked with 4% BSA blocking buffer in PBS for 1 h at RT. Cells were then rinsed with PBS and incubated with an anti-HA antibody (Cell Signaling) for 1 h at RT (dilution 1:100). Following rinsing with PBS, cells were washed with 0.1% Triton X-100 in PBS. Cells were then incubated for a further 1 h in the presence of secondary antibodies, Alexa Fluor 568 (Thermo Fisher Scientific), dilution 1:1000. Again, cells were rinsed once with 0.1% Triton X-100 in PBS and washed three times with PBS. Cells were then incubated with 1 μg/ml DAPI (Thermo Fisher Scientific) in PBS for 30 min at RT. Cells were rinsed three times and kept at 4 °C in PBS until ready to image using the fluorescence microscope. Measurement of HA intensity in the peripheral region around the nucleus and the nuclei was used to calculate a ratio between the periphery/nucleus. Paired Student t-test was used for statistical evaluation.

## Histopathology
Representative samples from each of the seven lung lobes were collected from each animal in accordance with the approved protocols by the NCI and/or NIAID Animal Care and Use Committees (ACUC; Protocol number: VB043) and preserved in 10% neutral buffered formalin for a minimum of 21 days. Subsequently, for all animals, one section of lung tissue from each of the seven lobes underwent processing using an automatic processor and was embedded into paraffin blocks. These blocks were then sectioned using a manual microtome and placed onto charged slides. Afterward, the slides were dried in an 80 °C oven for 1 h prior to Hematoxylin and Eosin (H&E) staining. The H&E staining process was carried out using the Sakura® Tissue-TekR Prisma™ automated Stainer, involving the application of commercial hematoxylin, clarifier, bluing reagent, and eosin-Y. A regressive staining method was employed, intentionally overstaining the tissues, and then utilizing a differentiation step (clarifier/bluing reagents) to remove excess stain. Finally, the slides were cover-slipped using the Sakura® Tissue-Tek™Glass® automatic cover slipper and allowed to dry before being evaluated in a blinded fashion for both control tissue and infected tissue.

## HTLV-1-*Gag* RNAscope
The presence of HTLV-1 virus was detected by staining 5 μm FFPE Rhesus macaque lung sections with the RNAScope® 2.5 LS probe V-HTLV-1-GAG (cat. #495058, ACD) with the RNAscope LS Multiplex Fluorescent Assay (cat. #322800, ACD) using the Bond RX auto-stainer (Leica Biosystems) with a tissue pretreatment 15 min at 95 °C with Bond Epitope Retrieval Solution 2 (Leica Biosystems), 15 min of Protease III (ACD) at 40 °C, and 1:750 dilution of OPAL™ 570 reagent (AKOYA, Biosciences®). The RNAscope 2.5 LS Negative Control Probe (Bacillus subtilis dihydrodipicolinate reductase (dapB) gene, cat. #312038) was used as a negative control. The RNAscope® LS 2.5 Positive Control Probe Mmu-PPIB (peptidylprolyl isomerase B) (cat#457718) encoding cyclophilin B was used as a technical control to ensure the RNA quality of tissue sections was suitable for staining. Slides were digitally imaged using a PhenoImager® HT 2.0 (AKOYA, Biosciences®). It is important to note that RNAscope, immunohistochemistry, as well as histopathology assays were performed on consecutive slices from the same tissue fragment.

## Immunohistochemistry
Immunohistochemistry staining was performed on LeicaBiosystems BondRX autostainer with the following conditions: Epitope Retrieval 1 (Citrate 20′), CD3 (cat. #MCA1477 Bio-Rad, 1:100 60′) with secondary antibody Rabbit anti-Rat IgG (Vector Laboratories), CD20 (cat. #M0755, DAKO/Agilent, 1:200 30′), Microglia/Iba1 (cat. #CP290, Biocare, 1:500 30′), Smooth Muscle Actin (cat. #ab5694, Abcam, 1:500 30′), and the Bond Polymer Refine Detection Kit (cat. #DS9800, LeicaBiosystems). Isotype control reagents were used in place of primary antibodies for the negative controls. Slides were removed from the

Bond autostainer, dehydrated through ethanols, cleared with xylenes, and cover-slipped.

## Viral score calculation

To assess viral infectivity of HTLV-1A and HTLV-1A/C$_{ol-L}$, four serological and virological assays (virus variables) measured at different time points were used to calculate timepoint-derived viral scores, variable-derived viral scores, and combination of those. All scores were then compared between the experimental treatment groups. Virus variables included viral DNA detection of gag (gag_PCR), viral DNA detection of orf-I/II (orf-I/II_PCR), number of bands in the HTLV serology assay, and HTLV-1 p24 antibody titer (p24_titer). Virus variables were normalized to 0–1 scale: specifically, gag_PCR and orf-I/II_PCR values were already binary (0/1), number_of_bands was divided by the maximum possible number of bands (=9), and $\log_{10}$(p24_titer) was divided by the maximum $\log_{10}$(p24_titer) value (= 4).

First, we calculated *Timepoint-derived scores* that combine the different virus variables for each animal at each time point. To give emphasis to the number of assays that showed a positivity for each time point and animal, the *fraction of variables that have any signal* was calculated. Finally, these 2 scores were combined by calculating the *combined timepoint-derived score* to merge the effects of the 2 previous scores.

(1). Timepoint-derived score for each timepoint and animal was calculated as the sum of all normalized virus variables divided by the total number of virus variables (=4), such that each tp_score$_{WeekXX,AnimalY}$ has a range of 0–1.

timepoints (=8), such that fracTPscores_notZero has a range of 0–1.

$$\text{fracTPscores\_notZero} = \frac{\sum_{i=Baseline}^{n=Week21} \text{tp\_comb\_score}_i \neq 0}{Total\,Number\,of\,Timepoints}$$

(6). The combination timepoint-derived score across all timepoints was then calculated as the average of the tp_comb_scores_Mean (from step 4) and the fracTPscores_notZero (from step 5)

$$\text{TP\_COMBO} = \frac{\text{tp\_comb\_scores\_Mean} + \text{fracTPscores\_notZero}}{2}$$

Next, we employed a different approach and calculated variable-derived viral scores individually for each virus variable. Similarly, in order to give emphasis to the total magnitude of the results of each assay, the *variable-derived score* for each animal was calculated by combining the different timepoints for each virus variable separately. To give emphasis to the number of assays that showed a positivity, the *fraction of timepoints that have any signal* was calculated for each assay and animal. Finally, these 2 scores were combined by calculating the *combined variable-derived score* to merge the effects of the 2 previous scores.

(7). Variable-derived score for each variable and animal was calculated as the sum of all normalized virus variables divided

$$\text{tp\_score}_{Week\,XX,\,AnimalY}$$
$$= \frac{\text{gag\_PCR}_{normWeekXX,\,AnimalY} + \text{pX\_PCR}_{normWeekXX,\,AnimalY} + \text{number\_of\_bands}_{norm\,Week\,XX,\,AnimalY} + \log_{10}(p24\_titer)_{norm\,Week\,XX,\,AnimalY}}{Total\,Number\,of\,Virus\,Variables}$$

(2). The fraction of variables that have any signal was calculated as the ratio of the number of virus variables not equal to zero divided by the total number of virus variables (=4), such that each fracVars_notZero$_{WeekXX,AnimalY}$ has a range of 0–1.

$$\text{fracVars\_notZeroWeekXX, AnimalY}$$
$$= \frac{\text{Number of Variables\_Not Zero}_{WeekXX},\, AnimalY}{Total\,Number\,of\,virus\,Variables}$$

(3). The combined timepoint-derived score for each timepoint/animal was calculated as the average of the tp_score$_{WeekXX,AnimalY}$ from step 1, and fracVars_notZero$_{WeekXX,AnimalY}$ from step 2. As above, each tp_comb_score$_{WeekXX,AnimalY}$ has a range of 0–1.

$$\text{tp\_comb\_score}_{WeekXX,\,AnimalY}$$
$$= \frac{\text{tp\_score}_{WeekXX},\,AnimalY + \text{fracVars\_notZero}_{WeekXX,\,AnimalY}}{2}$$

Then, for each animal, timepoint-derived scores for each timepoint were combined to get a set of composite timepoint-derived scores across all timepoints.

(4). tp_comb_scores (from step 3) were averaged across timepoints, such that tp_comb_scores_Mean has a range of 0–1.

$$\text{tp\_comb\_scores\_Mean} = \frac{\sum_{i=Baseline}^{n=Week21} \text{tpcombscore}_i}{Total\,Number\,of\,Timepoints}$$

(5). The fraction of tp_comb_scores (from step 3) with any signal was calculated as the ratio of the number of timepoints with tp_comb_score not equal to zero divided by the total number of

by the total number of timepoints (=8), such that each var_score$_{variableZ,AnimalY}$ has a range of 0–1.

$$\text{var\_score}_{variablrZ,\,AnimalY} =$$
$$\frac{\text{variableZ}_{normBaseline,\,AnimalY} + \text{variableZ}_{normWeek03,\,AnimalY} + \cdots + \text{variableZ}_{normWeek21,\,AnimalY}}{Total\,Number\,of\,Timepoints}$$

(8). The fraction of timepoints that have any signal was calculated as the ratio of the number of timepoints not equal to zero divided by the total number of timepoints (=8), such that each fracTPs_notZero$_{variableZ,AnimalY}$ has a range of 0–1.

$$\text{fracTPs\_notZero}_{variableZ,\,AnimalY}$$
$$= \frac{\text{Number of Time points\_NotZero}_{variableZ,\,AnimalY}}{Total\,Number\,of\,Timepoints}$$

(9). The combined variable-derived score for each variable/animal was calculated as the average of the var_score$_{variableZ,AnimalY}$ from step 7. and fracTPs_notZero_variableZAnimalY from step 8. As above, each var_comb_score$_{variableZ,AnimalY}$ has a range of 0–1.

$$\text{var\_comb\_score}_{variableZ,\,AnimalY}$$
$$= \frac{\text{var\_score}_{variableZ,\,AnimalY} + \text{fracTPs\_notZero}_{variableZ,\,AnimalY}}{2}$$

Next, for each animal, variable-derived scores for each variable were combined to get a set of composite variable-derived scores across all variables.

(10). var_comb_scores (from step 9) were averaged across variables, such that var_comb_scores_Mean has a range of 0–1.

$$\text{tp\_comb\_scores\_Mean} = \frac{\sum_{i=Baseline}^{n=Week21} \text{tp\_comb\_score}_i}{Total\,Number\,of\,Timepoints}$$

(11). The fraction of var_comb_scores (from step 9) with any signal was calculated as the ratio of the number of variables with var_comb_score not equal to zero divided by the total number of variables (=4), such that fracVARscores_notZero has a range of 0–1.

$$\text{fracVARscores\_notZero} = \frac{\sum_{i=\text{gag\_PCR}}^{n=\text{p24\_titer}} \text{var\_comb\_score\_NotZero}_i}{Total\,Number\,of\,Timepoints}$$

(12). The combination variable-derived score across all variables was then calculated as the average of the var_comb_scores_Mean (from step 10) and the fracVARscores_notZero (from step 11)

$$\text{VAR\_COMBO} = \frac{\text{Var\_comb\_scores\_Mean} + \text{fracVARscores\_notZero}}{2}$$

Finally, for each animal, composite scores combining timepoint-derived and variable-derived scores were calculated.

(13). Mean timepoint-derived combined score (from step 4) and mean variable-derived combined score (from step 10) were averaged.

$$\text{viral\_score\_Mean}$$
$$= \frac{\text{tp\_comb\_scores\_Mean} + \text{var\_comb\_scores\_Mean}}{2}$$

(14). Fraction not zero timepoint-derived combined score (from step 5) and fraction not zero variable-derived combined score (from step 11) were averaged.

$$\text{viral\_score\_fracScores\_notZero}$$
$$= \frac{\text{fracTPscores\_notZero} + \text{fracVARscores\_notZero}}{2}$$

(15). Combination timepoint-derived score (from step 6) and combination variable-derived score (from step 12) were averaged.

$$\text{viral\_score\_COMBO} = \frac{\text{TP\_COMBO} + \text{VAR\_COMBO}}{2}$$

### Other analytical methods

Significant changes in cell population frequency relative to baseline were modeled by fitting generalized estimating equations, as implemented by the geeglm function from the geepack R package. Mann–Whitney/Wilcoxon tests were used to determine differences between cell population levels or fold-changes between experimental groups. Heatmaps were generated using the pheatmap R package, and alluvia were manually drawn to guide the eye through the data. Full code can be found at https://github.com/NCI-VB/franchini_HTLVchimera_sarkis. Since our research design was hypothesis-generating, exploratory research, all p values are reported as nominal values without adjusting for multiple comparisons.

### Reporting summary

Further information on research design is available in the Nature Portfolio Reporting Summary linked to this article.

## Data availability

The sequence of the pAB_HTLV-1A/C$_{oI-L}$ chimeric molecular clone has been deposited in the NCBI GenBank nucleotide database under the accession no. PP860917. The HTLV-1C viral nucleotide sequence used in the cloning was derived from a virus isolated from PBMCs of an infected donor obtained from the Laboratory of Dr. Damian F.J. Purcell (accession nos. PP596271, PP596272, PP596273, PP596274, PP596275, PP596276, PP596277, PP596278, PP596279, PP596280, PP596281,

PP596282, PP596283, PP596284, PP596285, PP596286, PP596287, PP596288, PP596289, PP596290, PP596291, PP596292 for all patient proviruses). The data generated in this study have been deposited in Zenodo under accession code https://doi.org/10.5281/zenodo.16755081 https://zenodo.org/uploads/16755081[67]. Source data are provided with this paper.

## Code availability

The source codes for Figs. 3, 4 and 6, Supplementary Figs. 2, 4 and 6–9 are available at https://github.com/NCI-VB/franchini_HTLVchimera_sarkis.

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

## Acknowledgements

The authors thank David Ahern for editorial assistance. The authors gratefully acknowledge Aian Neil S. Alilil, Michelle Metrinko, and the staff at the NCI animal facility for expert care of rhesus macaque husbandry and technical assistance. The authors thank Hyoyoung Choo-Wosoba for randomization of the animal groups. The authors thank Katherine McKinnon (Vaccine Branch, NCI) for flow cytometry support. The authors thank Karim Bakhtiar and his team (Molecular Histopathology Laboratory, Fredrick, MD) for the histopathology, the RNAscope, and the Immunohistochemistry support. This research was supported by the Intramural Research Program of the National Institutes of Health (NIH). The contributions of the NIH authors were made as part of their official duties as NIH federal employees, are in compliance with agency policy requirements, and are considered Works of the United States Government. However, the findings and conclusions presented in this paper are those of the authors and do not necessarily reflect the views of the NIH or the U.S. Department of Health and Human Services.

## Author contributions

S.S. and G.F. conceived the study and wrote the paper, with contribution from all authors; S.S. constructed and characterized the chimeric molecular clone and performed the in vitro studies; S.S. coordinated, managed and led the animal study with the contribution of A.G.; S.S. and A.G. performed the whole blood phenotyping; M.A.R. performed the BAL and blood immune cell phenotyping and analyzed the data; L.S. performed the proximity extension assay; K.C.G. performed statistical analysis, visualization, and interpretation of the data; M.B. and S.S. performed the efferocytosis assay; C.A.P.-M. established the Jurkat and the THP-1 stably transduced cell lines with the assistance of S.B.; S.S., A.G., and R.W.-P. performed the HTLV p19 antigen ELISA; S.S. performed the HTLV-1 p24 antibody titer and the HTLV serology; F.E., A.W., and C.R. performed the histopathology, the immunohistochemistry and the RNAscope assays; R.M. performed the immunofluorescence assay; J.K., M.W.B., and K.K. performed surgeries and tissue collection; S.S., A.G., R.M., I.S.d.C., and M.D. processed the animal samples; D.F.J.P. provided the HTLV-1C patient sequences; Y.J. and L.S. performed antibody assay.

## Competing interests

The authors declare no competing interests.

## Additional information

[1]Animal Models and Retroviral Vaccines Section, Basic Research Laboratory, Center for Cancer Research, National Cancer Institute, Bethesda, MD, USA. [2]Advanced Biomedical Computational Science, Frederick National Laboratory for Cancer Research, Frederick, MD, USA. [3]Center for Cancer Research Collaborative Bioinformatics Resource, National Cancer Institute, Bethesda, MD, USA. [4]Department of Cell and Molecular Biology, Center for Immunology and Microbial Research, Cancer Center and Research Institute, University of Mississippi Medical Center, Jackson, MS, USA. [5]Molecular Histopathology Laboratory, Frederick National Laboratory for Cancer Research, Frederick, MD, USA. [6]Laboratory Animal Sciences Program, Frederick National Laboratory for Cancer Research, Frederick, MD, USA. [7]Clemson University, Clemson, SC, USA. [8]Vaccine Research Center, National Institute of Allergy and Infectious Diseases, National Institutes of Health, Bethesda, MD, USA. [9]Department of Microbiology and Immunology, The Peter Doherty Institute for Infection and Immunity, The University of Melbourne, Parkville, VIC, Australia. [10]Office of Scientific Programs, Center for Cancer Research, National Cancer Institute, Bethesda, MD, USA. ✉e-mail: sarkis.sarkis@hotmail.com; franchig@mail.nih.gov

