## [Peer Review file · Nature Communications]

High expression of Rex-orf-I and HBZ mRNAs and bronchiectasis in lung of HTLV-1A/C infected macaques

Corresponding Author: Dr Genoveffa Franchini

Version 0:

Reviewer comments:

Reviewer #1

(Remarks to the Author)

In this study, Sarkis et al. establish a robust macaque model to study HTLV-1C in vivo. Compared to the widely distributed HTLV-1A, HTLV-1C is endemic in Central Australia and is associated with respiratory failure and lung disease. Yet, neither a robust animal model nor the molecular cause of HTLV-1C-induced lung damage have been studied. Here, Sarkis et al. generate a chimeric virus of HTLV-1A (LTR, gag, pol, env) and HTLV-1C (pX region, 3'LTR) and comprehensively compare this chimeric virus to HTLV-1A in several cell lines and primary cells in vitro/ ex vivo and in the macaque model in vivo, the closest animal model system of HTLV-1-infection. To robustly infect the macaques, the authors perform a triple depletion of CD8+ T-cells, NK cells, and monocytes as previously established by the same working group. Viral replication and infectivity was comparable between both viruses in vitro and in vivo. A systematic analysis of immune cell populations in the blood and the BAL over time revealed a predominance of pro-inflammatory myeloid cells and neutrophils in animals infected with the chimeric HTLV-1A/C compared to HTLV-1A. Using proximity extension assay on plasma and BAL samples, the authors observe significant differences in cytokine profiles over time between HTLV-1A and HTLV-1A/C-infected animals. They correlate the cytokine profiles with all cell populations measured and find amongst others high levels of IL-15 in blood, which is a prognostic factor for pulmonary disease. In line with the cytokine profiles, the authors detect histopathological signs of lung disease like peripheral fibrosis or interstitial pneumonia in two triple-depleted animals upon HTLV-1A/C-infection. A closer analysis of viral gene expression reveals that HTLV-1A/C expresses a fusion protein called p16, which is related to p12 encoded by HTLV-1A. Compared to HTLV-1A, HTLV-1C lacks an AUG start codon at ORF1, but the first exon of rex in frame with ORF1 of HTLV-1A/C express the novel fusion protein p16. Functional analysis reveals that p16C from HTLV-1C, but not p12 encoded by HTLV-1A inhibits engulfment of cells by monocytes (efferocytosis). Finally, the authors also detect viral RNA in the lungs of infected animals using RNAscope, and they find an accumulation of immune cells surrounding fibrotic sections.

This is a very interesting, sound and comprehensive study and the first to develop a robust non-human primate model to monitor HTLV-1C replication and its association with HTLV-1C-induced lung disease. Beyond, this study also shows that despite lacking a start codon – HTLV-1C has the capacity to encode an ORF1-like protein, p16C, which blocks efferocytosis and may thus contribute to HTLV-1C-induced lung disease. Together, this work is of very high significance to this and related fields. The work shown supports the conclusion, however, I have some comments to be considered by the authors.

1. Why did the authors not use wildtype HTLV-1C but a fusion of HTLV-1A and HTLV-1C? Did wildtype HTLV-1C not express/ produce virus? This is very important to know for the reader and should be explained. Despite conserved splice sites and similar viral transactivation as shown in this study, there might be differences between the chimeric HTLV-1A/C and HTLV-1C.
2. The authors use triple depleted animals to robustly infect the macaques with HTLV-1A/C and the control virus HTLV-1A. How can the authors exclude that the initial triple depletion is partially responsible for the observed clinical signs and cytokine profiles? Maybe non-depleted animals would not develop the clinical signs and would also exhibit a different cytokine and cell population profile. Please discuss the limitations of initially "immunocompromised" model.
3. The authors showed histopathological findings from lungs of HTLV-1A/C-infected animals, but not from HTLV-1A since this was not available from the same time point. However, this would be very important, if not from the same time point, from

a related time point at all, to substantiate the observed findings that HTLV-1A is not causing similar lesions in lungs of infected animals. Please provide more information on pathologies usually observed upon HTLV-1A infection of macaques.

4. Figure 5: The finding that HTLV1-A/C expresses a Rex-ORF1 fusion protein is very interesting and the data showing that HTLV-1A/C p16C inhibits efferocytosis, but HTLV-1A p12 does not is important. Is there experimental evidence how this works, e.g. does p16C block LOX-1 expression compared to p12?

Minor:

1. Figure 1: Authors should enhance the frequency of CBLs to be tested. (n=1)
2. Figure 1c-e: Authors use t-test (d) and two-tailed Mann-Whitney test (d,e,f), however, more than two groups are compared. Please use another test.
3. Supplementary Figure 2A: The study design should be shown in the main paper.
4. Figure 2: Please change label and colors of "Triple depletion HTLV-1A/C" and "Triple depletion HTLV-1A" – it is very difficult to see the different colors.
5. Figure 3: This figure is well-designed to show a highly complex data set. However, for some readers (e.g. color blind), it might be useful to show the data as bar charts as well (e.g. in the supplement).
6. Figures 5 c/e and d/f show the same data for the control Jurkat-GFP. This should either be mentioned in the legend or data should be presented in one graph ((c+e) and (d+f)).

Reviewer #2

(Remarks to the Author)

• What are the noteworthy results?

This manuscript investigates in animals the molecular mechanisms responsible of the pathology associated to the HTLV-1C infection, which in human is linked to severe lung diseases, in contrast to HTLV-1A infection which barely causes lung symptoms but leukemia or neurodegeneration. Sequences analyses revealed that the main divergent regions between HTLV-1A and HTLV-1C are located in the 3' sequences of the HTLV-1 genomes, in particular within the accessory/regulatory sequences. This region is responsible for the expression of proteins reported to play crucial roles in HTLV-1A infection and pathogenesis such as Tax, HBZ, p30, or p8/12 proteins. The role of HTLV-1C accessory/regulatory proteins is currently not known. To investigate this, the authors used an elegant strategy in which they created a chimeric virus bearing the structural genes of HTLV-1A fused to the accessory/regulatory sequences of HTLV-1C. This leads to a chimeric clone allowing expression of infectious virions in vitro. Due to the lack of cross-reactivity antibodies, the expression of accessory/regulatory proteins could not be controlled.

The chimeric clone was then used to infect macaques, that were depleted in CD8+ T-cells, NK cells and monocytes to support chronic infection. The results of the number of infected cells and the breath of seroconversion were compared to animals infected with HTLV-1A in the same condition, from a previous study performed by the authors and published previously. They concluded to similar in vivo infectivity, but different immune cell counts in blood or bronchoalveolar lavages in HTLV-1A/C chimeric and in HTLV-1A-infected animals. Specifically, they observed in HTLV-1A/C chimeric animals more inflammatory infiltrated dendritic cells in the early weeks after infection followed by inflammatory neutrophils later, suggesting lung disease, with lung sign of pathology confirmed in some animals by lung histopathology after necropsies. While the demonstration missed a formal description of symptomatology relative to lung disease for reader not familiar to the physiopathology of this disease (specifically, how "sick" these HTLV-1A/C animals are compared to "healthy" non infected macaques?), this part of the work is convincing and supported by a huge amount of data: several animals were used, the longitudinal analysis of immune cells was performed longitudinally. It demonstrates a different course of infection when HTLV-1C 3' sequences are expressed in a HTLV-1A background.

The second part of the work aims at demonstrating that the protein p16C, expressed from a rex-orf-I fusion is responsible of the lung manifestation correlated to HTLV-1A/C infection. This is less convincing, in particular because the authors can not exclude the involvement of other proteins (Tax, HBZ or P30) expressed from the same sequences. Moreover, the inflammation of lung described after histopathological analyses is modest and no demonstration of juxtaposition of infiltrated inflammatory immune cells in proximity to infected cells is provided.

Therefore, the statement that "p16C" is linked to lung disease" must be deemphasized in the title and the abstract, as the role of efferocytosis in the lung disease not demonstrated in the infected animals.

Below are specific major comments to address before considering publication.

• Will the work be of significance to the field and related fields? How does it compare to the established literature? If the work is not original, please provide relevant references.

YES after corrections and de emphasized interpretation.

• Are there any flaws in the data analysis, interpretation and conclusions? Do these prohibit publication or require revision? YES figure 3 and related sup fig 5 and 6 are difficult to read due to a huge amount of information that are not fully and exhaustively commented in the text. Simplification and focus interpretation must be performed. The formal demonstration of p16C involved in the induction of lung disease during HTLV-1A/C infection is not provided.

• Is the methodology sound? Does the work meet the expected standards in your field? YES

• Is there enough detail provided in the methods for the work to be reproduced? YES

• Does the work support the conclusions and claims, or is additional evidence needed? NO

The title "HTLV-1C lung disease linked to p16 protein inhibition of efferocytosis in macaques" is overstated and not fairly supported by the results presented in the text for several reasons:

- lung disease is not formally demonstrated in macaques as no clinical symptoms are reported during longitudinal follow-up post infection. Higher infiltration of inflammatory dendritic cells and neutrophils in broncho-alveolar lavages in HTLV-1A/C infected macaques compared to HTLV-1A may not be linked to objective disease. Please provide clinical analysis of the infected macaques with objective sign of lung disease. What are the number of inflammatory infiltrated cells and how those correlates with known lung diseases? Same comment for the level of cytokines detected in blood and in BAL. Histology of the lung at necropsy show different patterns in the 3 animals analyzed those seems modest (with rare infiltration of immune cells or low peripheral fibrosis). Comparison with normal lung would help to appreciate the disease state. Are the infiltrated cells shown in figure 6c localized /in close proximity to infected cells shown in a and b panels?
- p16 inhibition of efferocytosis is shown in vitro after transduction of recombinant p16-HA (originated from a rex-orf-I construct derived from HTLV-1C genome) in jurkat as target cells and human CD14+ cells or human monocytic cell lines as effector. Despite rex-orf1 mRNA is detected in 729-6 lymphoblastic B cell line infected by HTLV1A/C chimera and this synthetic construct allow expression of a p16 protein after transduction of jurkat cells, it does not prove that the same mRNA will allow expression of a p16 protein in infected macaques and that this protein is responsible for lung disease and inhibition of efferocytosis. Such formal demonstration would require infection of macaques with HTLV-1A/C bearing a KO of p16. Such experiment are obviously difficult to perform for a revision, thus the title and the manuscript should be write with more caution to avoid over interpretation of data.

What are the evidence of HTLV-1C orf-I expression in HTLV-1C infected cells (in vitro after transfection of the molecular clone or derived from patients)? Are there in vitro data showing the presence of mRNA in HTLV-1C infected cells (lines or derived from patients?). The presence of rex-orf-1 mRNA from the chimeric HTLV-1A/C molecular clone is not sufficient to state that orf-I protein from HTLV1C, although similar to the orf-I from HTLV1A (ie the p12/8 proteins), is expressed and has a function in infectivity and/or disease.

The role of other proteins (Tax, HBZ and p30) expressed from the orf-I-IV-LTR region of HTLV-1C genome is not addressed and as mentioned by the authors in the discussion, they might be involved in HTLV-1C and HTLV-1A/C chimera pathogenesis. Tax expression from HTLV-1A/C chimera is not detected in WB using the TAB172 antibody. Would another clone (ie LT4 or 1A3) be more efficient? Although Tax must be expressed since NFkB and LTR are induced, its level of expression might differ from that expressed by HTLV-1A and could be involved in HTLV-1A/C pathology in macaques. Similarly, the level of HBZ protein in HTLV1-A/C infected animals would be important to evaluate in comparison to that of HTLV-1A infected animals to determine whether other viral protein could be involved in the different immunophenotype observed in this study. Specifically, because authors state expression of p12 from HTLV-1A alone is not sufficient to block efferocytosis in vitro, while they demonstrated the opposite in a previous paper. The author state that this suggesting that other viral proteins could be necessary to control efferocytosis and pathogenesis of HTLV-1A infection. The same might hold true for p16, those expression at the protein level in HTLV-1A/C infected animals is not demonstrated.

Minor:

The results regarding the different populations of monocytes presented in figure 3b is not commented in the text.

Figure 3 summarizes a huge amount of data and is difficult to read. Perhaps consider simplification with highlight of striking results (such as the level of pDC/mDC/neutrophils in blood or BAL) presented as histograms.

The sentence line 178 is not fully correct: the level of IL8+mDC and IN and NC IL8+monocytes in blood at w12 is lower in HTLV-1A/C infected animals than HTLV-1A. They cannot mirror that of IL8+neutrophils because this cell population is not shown at week 12. In the same line, how the authors explain that proinflammatory (expressing IL8 or TNFa) mDC and pDC are not found in BAL of HTLV-1A/C infected animals at week 12 and 21? is the inflamed lung only supported by infiltration of CD64+ neutrophil in BAL and production of CCL13 cytokine?

There is no molecular weight in the WB in sup figure 8c and 8d rending difficult to identify p8 vs p12 that is claimed to be present in THP1 but not in JK. Several bands are present in THP1 expressing p16C that are not present in JK, what is the explanation?

The GFP signal detected in figure 5b is not commented, as the experimental setting is not precise in the text nor in the figure legend. Reader understands that cells were transduced with a lentiviral vector expressing GFP that is used to control/identify transduced cells after careful reading t-of the methods. This must be correct to ease the comprehension.

Reviewer #3

(Remarks to the Author)

HTLV-1C lung disease linked to p16 protein inhibition of efferocytosis in macaques

Background: HTLV-1 A is the predominant subtype whereas HTLV-1 C is rare and is known to cause respiratory failure and premature death. To better understand the different pathology associated with the 2 subtypes, Sarkis Sarkis and colleagues developed a HTLV-1A/C chimeric to recapitulate infection with HTLV-1A with the pathology of HTLV-1C in a macaque model.

The authors conclude that HTLV-1A/C induced lung pathology as reported in humans and accompanied with high infiltration of pro-inflammatory innate effector cells. In contrast, HTLV-1 A induced the resolution of inflammation accompanied by higher expression of IL10. Sarkis et al. identified a 16kDa fusion protein expressed by HTLV-1A/C which reduced clearance of infected T cells by macrophages/monocytes and thus providing a target for HTLV-1 C mediated lung pathology. Overall, the conclusions of the authors are not fully supported by the data or the presentation of the data. In detail:

Major comments:

Is there any evidence that the protein P16C is expressed in humans and is associated with lung pathology?

It is unclear whether HTLV-1A and HTLV1A/C show the same infectivity and virus replication in vivo. The two virus types were not compared head-to-head in the same experiment. It is hard to judge if the statement "Infectivity of HTLV-1A and chimeric HTLV-1A/C was equivalent in macaques" is correct given this major limitation. The viral scores and relative units in Figure 2 are difficult for a reader to understand and evaluate. Viral titers in serum, PBMCs and potentially other compartments such as the BAL should have been reported with time kinetics and clinical scores.

Figure 3 is very complex and not well explained. No gating strategy for flow cytometry (only as text). What was the frequency of IL10+ or TNFa+ cells straight ex vivo? Was the frequency robust enough for downstream analysis? For example: How many events of pDCs expressing IL10 were collected which allow robust analysis of such sub-populations? FACS plots of MPO staining including control staining? No tracking of individual cell populations in total numbers and frequencies side-by-side in order to judge whether innate immune cell numbers in BAL or blood are truly and significantly altered between the virus variants. Why was week 5 chosen as the first time point?

Given that in this model, 3 major cell populations (CD8 T cells, NK cells and monocytes) are depleted for some period of time it is difficult to exclude that this has an impact on subsequent cell dynamics and cytokine production of all other cell types. Further, different or heterogeneous cell discovery rates between animals and groups might (for example Suppl Figure 4b and c) make these dynamics even more complex. This is a limitation of the immunological investigations at later time points as reported in Figure 3 and 4.

The lung of HTLV-1A was not investigated for lung pathology and lung pathology for the HTLV-1A/C infected animals was not quantified. Only representative images are shown. Thus, the statement: "suggesting causal association between HTLV-1A/Col-L infection and lung disease" is to some extent overinterpretation or speculation.

P16C staining is not shown in Figure 6 and is not clear whether this is highly expressed in the lung of HTLV1-A/C infected animals. No comparison to HTLV-1A animals is available to judge whether CD3 T cell clearance is prevented in HTLV-1A/C infected animals.

Minor:

High prevalence of HTLV-1C in Aboriginal populations might point towards a human genetic component? Please discuss 1G: Difference in virus replication at week 1 post infection in vitro significant? This earlier time point is missing for 1H and 1I. Why did the authors look only at time points 2 weeks post co-culture?

Figure 5c-f: Efferocytosis rates appear overall quite low. Did the authors try to increase the ratio of Jurkat or CD4 T cells to monocytes to increase the overall efferocytosis rate?

Version 1:

Reviewer comments:

Reviewer #1

(Remarks to the Author)

In their revised version, the authors have addressed all of my points of criticism.

In contrast to HTLV-1A/C-infected macaques, the authors find no signs of bronchiectasis in the newly collected data from HTLV-1A-infected macaques, which supports their initial hypothesis. In addition to Rex-orfl, the authors now extended their analysis and also detect increased expression of HBZ mRNA in the lungs of HTLV-1 A/C-infected animals.

Reviewer #2

(Remarks to the Author)

The authors greatly improved the manuscript and replied convincingly to requests. The paper is very complete and deserves publication

Reviewer #3

(Remarks to the Author)

The revised manuscript by Sarkis et al. provides additional data, analysis and conclusions. The authors responded to all questions/concerns raised by the reviewer. The study was truly improved by an additional study/data for the HTLV-1A group for comparison. However, the comparison of HTLV-1A to HTLV-1A/C appears to show only moderate - small difference in the NHPs between HTLV-1A and HTLV-1A/C as presented. Overall, in case the authors would argue that major differences exist, they are not well visualized or analyzed. I focused primarily on the analysis in Figure 3 accompanied by Suppl Figures and raw data tables provided for the revised manuscript. In general it was very difficult to understand the plots, analysis and how this was connected to the conclusions in the text. It took several hours to get some good idea about the nature of the

data, the legends, labeling, analysis and interpretation. Details of my major comments are below:

- The gating strategy in Suppl Figure 5 appears incomplete. Is this PBMCs or BAL? Which Animal and time point? In the plot titled "Non-T, B and NK cells: Why do most cells appear as CD8+CD3+CD20-A700? Where are the CD4 T cells and B cells? Gated out before? Its unclear.
- In case this is a blood sample: In the gate Macrophage/monocytes: The majority of cells appear to be macrophages. They should not really exist at all in blood (or below 0.1%). Macrophages reside / differentiate in tissue.
- Having said that, the frequency of macrophages appears very low in the BAL (here the frequencies are generally reported with 80-90%). Whether this is an effect of the triple depletion is not clear. However, it does not seem to be different 21 weeks post infection or at the earlier time points (raw data table).
- Classical monocytes appear to be at 28.9% (Suppl gating strategy). This seems low considering they should make the vast majority of all monocytes (in humans). The ref 67 provided in the method section for NHPs monocyte subpopulations does not provide this information. In fact, in that paper I did not find any analysis of monocytes.
- Given my concern regarding the macrophage population it is not clear from the information provided that the gating strategy identifies the majority of myeloid cells correctly. Further the gating for mDCs does not appear to show a distinct population. In line with this it is surprising that 14% of mDCs appear to be MPO+ (reviewer comments) considering it is a bona fide neutrophil marker.
- There are no flow plots showing the staining/gating for IL8, TNF α and IL10. The frequency of myeloid cells expressing these chemokine/cytokines straight ex vivo appear very high (up to 99%).
- Thus, the conclusions from this analysis appear not convincing at all. Looking at the tables provided as raw data, the gating strategy and the missing information all together it appears the differences described in the text (line 185-212) are not supported by the data and the analysis.

2 minor/other points:

For the proteomics analysis of chemokines and cytokines the analysis as provided does not allow to judge whether there are indeed differences. A table with the median of each analyte in each group and an appropriate statistical test (non-parametric corrected for multiple comparisons) should be provided along with boxplots or bargraphs for each analyte. This would help the reader to fully appreciate any differences.

The differences in pathology and viral load in the lung appear minor.

Version 2:

Reviewer comments:

Reviewer #3

(Remarks to the Author)

Dear authors, thank you very much for the revised manuscript and for answering/clarifying my comments/concerns. There are a few points which remain unclear even after revision. This is primarily true for the flow cytometry analysis. Please find specific comments in the word file.

(Remarks on code availability)

Version 3:

Reviewer comments:

Reviewer #3

(Remarks to the Author)

Dear authors,
thank you very much for the clarifications. Thank you for providing the FMO data. Since the high ex vivo expression of cytokines in myeloid cells appears unusual it would be informative to include the FMOs along the gating strategy so that readers can evaluate it as well.

(Remarks on code availability)

REVIEWER COMMENTS

Reviewer #1 (Remarks to the Author):

Major comments

1. Why did the authors not use wildtype HTLV-1C but a fusion of HTLV-1A and HTLV-1C? Did wildtype HTLV-1C not express/ produce virus? This is very important to know for the reader and should be explained. Despite conserved splice sites and similar viral transactivation as shown in this study, there might be differences between the chimeric HTLV-1A/C and HTLV-1C.

Primary samples from Aboriginal people living with HTLV-1C were not available due to strict regulations affecting the export of blood from HTLV-1C infected individuals to the United States. We therefore elected to generate *in silico* chimeric clones of HTLV-1C virus, encompassing either only the type C 3' end and the viral LTR (HTLV-1A/*C_{ol-L}*) or part of the *pol*, the entire *env*, and the 3' end regulatory genes (HTLV-1C_{*E-L*}). We demonstrate in Supplementary **Fig. 10d-f** that the splicing pattern of *orf-I* in HTLV-1A/*C_{E-L}* and HTLV-1A/*C_{ol-L}* chimeras did not differ. For more clarity and as suggested by the reviewer the main text was modified (lines 80-82, 321-323)

2. The authors use triple depleted animals to robustly infect the macaques with HTLV-1A/C and the control virus HTLV-1A. How can the authors exclude that the initial triple depletion is partially responsible for the observed clinical signs and cytokine profiles?. Maybe non-depleted animals would not develop the clinical signs...

Given that we see the same pathology in non-depleted and triple depleted animals infected with HTLV-1A/ *C_{ol-L}*, it is unlikely that triple depletion contributes to the clinical signs and cytokine profile. Histopathological analyses were performed on three macaques inoculated with the HTLV-1A/*C_{ol-L}* chimeric virus. TiT, a replete animal and TMN and DG8Z subjected to triple depletion of CD8⁺ T cells, NK cells, and monocytes prior to inoculation. Despite the lack of triple depletion, animal TiT exhibited pulmonary lesions, including bronchiectasis, pulmonary hemorrhage, edema, and focal pneumonia (**Figure 5a, right panel; Supplementary Table 2** (lines 283-288).

...and would also exhibit a different cytokine and cell population profile. Please discuss the limitations of initially “immunocompromised” model

The impact of triple depletion on immune responses and cytokine/chemokine profiles in both blood and bronchoalveolar lavage (BAL) fluid were compared between the replete (n=4, HTLV-1A/*C_{ol-L}*) and triple-depleted (n=5, HTLV-1A/*C_{ol-L}*; n=10, HTLV-1A) animals at multiple time points (weeks 5, 12, and 21).

While significant increases in several cytokine/chemokine levels in triple depletion vs no depletion between experimental groups were observed in the blood at an early timepoint (week 5, post-depletion) as expected; (**Supplementary figure 8**) by later timepoints (weeks 12 and 21), after recovery of key immune cell populations such as monocytes (week 1) and NK cells (week 5) (**Supplementary Fig. 4**), the levels of these markers were comparable between triple depleted HTLV-1A/*C_{ol-L}* and replete HTLV-1A/*C_{ol-L}* groups (**Supplementary figure 8**).

In contrast, in the BAL compartment, fewer significant alterations in cytokine/chemokine levels between triple and non-depleted animals infected with HTLV-1A/*C_{ol-L}* were detected at the earlier

timepoint (week 5), with the majority of the differences observed, mostly decreases in the triple depleted group, only at the later time points (weeks 12 and 21) after cell recovery (**Supplementary figure 8b**). Importantly, most differences in cytokine/chemokine levels in BAL between triple and non-depleted animals were unique to either the HTLV-1A/C_{ol-L} triple depleted group or the HTLV-1A triple depleted group, relative to the HTLV-1A/C_{ol-L} non-depleted group. This suggests that virus type prevails as the primary differentiating variable in the cytokine/chemokine profiles.

We believe this demonstrates that while triple depletion may transiently affect cytokine/chemokine levels in the peripheral blood at early stages, it does not impact these markers at the site of pathology (lung) during the same timeframe, thereby suggesting that the triple depletion is not a key element for the onset of lung lesions. The possibility that the triple depletion accelerates the worsening of clinical symptoms by facilitating a more robust infection in macaques cannot be ruled out, albeit more robust infection was measured for both viruses. (Line 236-251)

3. The authors showed histopathological findings from lungs of HTLV-1A/C-infected animals, but not from HTLV-1A since this was not available from the same time point. However, this would be very important, if not from the same time point, from a related time point at all, to substantiate the observed findings that HTLV-1A is not causing similar lesions in lungs of infected animals. Please provide more information on pathologies usually observed upon HTLV-1A infection of macaques.

We recognize that this is an important point. Indeed, we have performed a Study 2 to investigate this, which we have now added to the manuscript (**Supplementary. Fig 3e-h**) (Line 147-157). We combined the results from Studies 1 and 2 and reanalyzed data on virus score, cytokines, and cellular responses in blood and lung together on 5 triple depleted, and 4 replete animals infected with HTLV-1A/C_{ol-L} (total n=9), 5 new and 5 historical triple-depleted animals infected with HTLV-1A (**Supplementary Fig. 3i**). We combined the current and historical data from animals infected with HTLV-1A (total n=10) since they had a similar systemic virus score, which remained similar to that of HTLV-1A/C_{ol-L} infected macaques (**Fig. 2d,e**). Inclusion of these new studies clearly demonstrate a difference in the virus score in lung between HTLV-1A and HTLV-1AC infected animals, as did lung pathology (**Fig. 6** and **Supplementary Table 2** and **3**). (Line 284-287)

4. Figure 5: The finding that HTLV1-A/C expresses a Rex-ORFI fusion protein is very interesting and the data showing that HTLV-1A/C p16C inhibits efferocytosis, but HTLV-1A p12 does not is important. Is there experimental evidence how this works, e.g. does p16C block LOX-1 expression compared to p12?

To address this question, we performed *in vitro* experiments on the monocytic THP-1 cell lines stably expressing either p16C or p12 following treatment with Phorbol 12-myristate 13-acetate (PMA) for 24 hours, and performed proteome analyses of supernatants. Protein levels were assessed using a Proximity Extension Assay (PEA), with eight biological replicates for each cell line (n=8). The proteomic data revealed that THP-1 cells constitutively expressing p12 exhibited significantly higher levels of Lectin-like oxidized low-density lipoprotein receptor-1 (LOX-1) in THP-1 cells expressing p16C (data not shown). The pro-resolution IL-10 cytokine that promote efferocytosis was decreased in THP-1 cells expressing p16C (**Supplementary Fig. 12c** (lines 351-356)).

This finding suggests that p12, is associated with higher expression of LOX-1, a protein implicated in protecting lung tissue by enhancing the clearance of apoptotic cells through efferocytosis, and that p16 C is linked to decreased expression of IL-10

Minor:

1. Figure 1: Authors should enhance the frequency of CBLs to be tested. (n=1)

We agree and in order to strengthen the data we perform an additional experiment using a additional cord blood from a new donor. Data on p19 Gag expression is reported in **Fig. 1g** (line 115)

2. Figure 1c-e: Authors use t-test (d) and two-tailed Mann-Whitney test (d,e,f), however, more than two groups are compared. Please use another test.

As suggested by the reviewer, and because more than two groups were compared, the two-tailed Mann-Whitney test was replaced with a Kruskal-Wallis corrected for multiple comparisons using the Dunn test.

3. Supplementary Figure 2A: The study design should be shown in the main paper.

As suggested by the reviewer, the study design previously presented in Supplementary Fig. 2a,b has been moved to main manuscript **Fig. 2a,b**.

4. Figure 2: Please change label and colors of “Triple depletion HTLV-1A/C” and “Triple depletion HTLV-1A” – it is very difficult to see the different colors.

We have changed the label and colors in **Fig. 2** as suggested.

5. Figure 3: This figure is well-designed to show a highly complex data set. However, for some readers (e.g. color blind), it might be useful to show the data as bar charts as well (e.g. in the supplement).

We have redesigned Figure 3 to contain scatterplots as suggested by the reviewer such that position of the points can be used rather than color alone to assess the differences in the datapoints. Updated **Supplementary Figures 6 and 7** contain additional representations of this data in heatmap format.

6. Figures 5 c/e and d/f show the same data for the control Jurkat-GFP. This should either be mentioned in the legend or data should be presented in one graph ((c+e) and (d+f)).

We created a new **Supplementary fig 12a,b** (previously mentioned as figure 5 c/e d/f) where as per the reviewer’s suggestion, we combined the specified panels into single graphs, illustrating that the same control was used for both p12 and p16.

Reviewer #2 (Remarks to the Author):

Major comments

1. Lung disease is not formally demonstrated in macaques as no clinical symptoms are reported during longitudinal follow-up post infection.

Infected animals were monitored daily by technical staff and veterinary staff for clinical signs. The veterinary staff report that pneumonia is often not identified until the disease has significantly progressed. This delayed recognition is due to animals' physiological ability to compensate for illness, which can obscure clinical signs until the disease reaches an advanced stage. Semiannually, animals undergo physical exams in which their lungs are auscultate and pulse oximetry is performed. No overt signs of disease were noticed in these animals but, as with the daily assessments, it is the veterinarian's experience that lung disease is not easily noticed when mild.

Higher infiltration of inflammatory dendritic cells and neutrophils in broncho-alveolar lavages in HTLV-1A/C infected macaques compared to HTLV-1A may not be linked to objective disease.

Please note that ~15% of humans infected by HTLV-1C develop lung disease by an average age of 40. Our animals were sacrificed at 10 months post infection, so severe disease was not expected. However, the finding that 2 of 3 animal had bronchiectasis suggests difference between the two viruses

Please provide clinical analysis of the infected macaques with objective sign of lung disease. What are the number of inflammatory infiltrated cells and how those correlates with known lung diseases?

Same comment for the level of cytokines detected in blood and in BAL.

We agree with the reviewer that we have not formally linked virus induced inflammation to lung disease. However, the inflammatory signature observed in HTLV-1A/C infection has been linked to lung diseases as well as defective efferocytosis in humans (Korkmaz, F.T., et al. Lectin-like oxidized low-density lipoprotein receptor 1 attenuates pneumonia-induced lung injury. JCI Insight 7(2022)). These data support the hypothesis put forward by our paper that the higher inflammatory profile induced by HTLV-1A/C may favor lung disease development.

Histology of the lung at necropsy show different patterns in the 3 animals analyzed those seems modest (with rare infiltration of immune cells or low peripheral fibrosis). Comparison with normal lung would help to appreciate the disease state.

We did not have normal lung tissue for these studies, however we did repeat the infection of animals with HTLV-1A. Among the 3 animals infected with HTLV-A, no bronchiectasis was detected. Among the 3 animals infected with HTLV-1A/C, 2 of 3 animals had bronchiectasis. These data are summarized in **Fig. 6d**. We cannot exclude that HTLV-1A may cause lung disease. Indeed in humans HTLV-1A very rarely causes lung disease versus the morbidity reported in the aboriginal people of approximately 15%.

Are the infiltrated cells shown in figure 6c localized /in close proximity to infected cells shown in a and b panels?

Staining analysis was done on consecutive block slices and indicates that infiltrated cells are in close proximity to infected cells. We have now indicated this in the material and methods section (Line 1358-1360).

Quantification of inflammatory infiltrated cells:

We report differences in the percentage of cells type infiltrating the lung between HTLV-1A and HTLV-1A/C, using BAL as a surrogate.

Normal lung histopathology slides:

Please see Supplementary Fig 9c

Clarification on IHC and RNAscope:

Could we confirm whether the immunohistochemistry (IHC) was performed on the same slides, and whether the infiltrated cells were in close proximity to gag RNAscope-positive cells? From the current data, it appears that this is indeed the case, but explicit confirmation would be helpful.

RNAscope and IHC were indeed performed on consecutive slices from the same tissue fragment. We have now indicated this in the material and methods section (Line 1358-1360).

Quantification of lung pathology:

The histopathology is multifocal, and we are unable to properly quantitate it in all lung lobes. Bronchiectasis can be easily identified.

2. p16 inhibition of efferocytosis is shown in vitro after transduction of recombinant p16-HA (originated from a rex-orf-I construct derived from HTLV-1C genome) in jurkat as target cells and human CD14+ cells or human monocytic cell lines as effector. Despite rex-orfI mRNA is detected in 729-6 lymphoblastic B cell line infected by HTLV1A/C chimera and this synthetic construct allow expression of a p16 protein after transduction of jurkat cells, it does not prove that the same mRNA will allow expression of a p16 protein in infected macaques and that this protein is responsible for lung disease and inhibition of efferocytosis. Such formal demonstration would require infection of macaques with HTLV-1A/C bearing a KO of p16. Such experiment are obviously difficult to perform for a revision, thus the title and the manuscript should be write with more caution to avoid over interpretation of data.

We modified the title and manuscript language as suggested and included HBZ since the spliced mRNA is found more frequently in the lung of HTLV-1A/C than HTLV-1A infected animals

3. What are the evidence of HTLV-1C orf-I expression in HTLV-1C infected cells (in vitro after transfection of the molecular clone or derived from patients)? Are there in vitro data showing the presence of mRNA in HTLV-1C infected cells (lines or derived from patients?). The presence of rex-orf-1 mRNA from the chimeric HTLV-1A/C molecular clone is not sufficient to state that orf-

I protein from HTLV1C, although similar to the orf-I from HTLV1A (ie the p12/8 proteins), is expressed and has a function in infectivity and/or disease.

Unfortunately, we do not have access to patient cells or lines derived from HTLV-1C infection. To further support expression of p16C, we evaluated also *rex-orf-I mRNA* expression in the context of a **novel HTLV-1A/C chimeric molecular clone**, HTLV-1A/C_{E-L}, encompassing most of *pol*, the complete *env*, *orf I, II, III, IV*, and the 3'LTR of HTLV-1C. Transcriptional analysis of the HTLV-1A/C_{E-L} chimeric virus following transient transfection in 293T cells demonstrated expression of all unspliced, singly spliced, and doubly spliced viral mRNAs, including *rex-orf-I* (**Supplementary Fig. 10 d,e**) (Line 321-324). While we do not have at present reagents to detect the p16C protein in infected cells, we provide evidence here that the *rex-orf-I mRNA* is expressed in the lung of infected animals. (Line 375-380).

4. The role of other proteins (Tax, HBZ and p30) expressed from the orf-I-IV-LTR region of HTLV-1C genome is not addressed and as mentioned by the authors in the discussion, they might be involved in HTLV-1C and HTLV-1A/C chimera pathogenesis.

To better address the contribution of additional viral proteins, RT-PCR was performed in the seven lung lobes of the animals infected by either HTLV-1A or HTLV-1A/C. We found a significantly higher frequency of expression of Gag, *rex-orf-I* and spliced HBZ mRNAs in the lung of HTLV-1A/C than HTLV-1A infected animals (**Fig 6. b-c**). Tax expression did not differ. (Line 375-380)

Tax expression from HTLV-1A/C chimera is not detected in WB using the TAB172 antibody. Would another clone (ie LT4 or 1A3) be more efficient?

We tested the 1A3 and the-LT4 antibodies as requested. The data are now shown in the **Supplementary Fig. 1b**).

Although Tax must be expressed since NFkB and LTR are induced, its level of expression might differ from that expressed by HTLV-1A and could be involved in HTLV-1A/C pathology in macaques. Similarly, the level of HBZ protein in HTLV1-A/C infected animals would be important to evaluate in comparison to that of HTLV-1A infected animals to determine whether other viral protein could be involved in the different immunophenotype observed in this study. Specifically, because authors state expression of p12 from HTLV-1A alone is not sufficient to block efferocytosis in vitro, while they demonstrated the opposite in a previous paper.

We wish to bring clarity to the reviewer's statement. The current result, obtained with p12 expression in isolation, does not conflict with the published results obtained with p12 expression in the context of the entire virus (compared with a p12 knock-out molecular clone). The published results show that p12 expression in context of the entire virus results in upregulation of the efferocytosis "don't eat me signal", while the current manuscript suggests the p12 expression in isolation may require the participation of additional viral gene(s), reflecting the complexity of the effect elicited by the expression of multiple viral genes on immunity. We clarified this point in the new version of the paper (Line 425-433).

The author state that this suggesting that other viral proteins could be necessary to control efferocytosis and pathogenesis of HTLV-1A infection. The same might hold true for p16, those expression at the protein level in HTLV-1A/C infected animals is not demonstrated

We agree with the reviewer that we cannot exclude also the contribution of other viral proteins in inhibition of efferocytosis in the context of the entire viral genome and we have clearly stated it in the manuscript (Line 430-433).

Minor:

1. The results regarding the different populations of monocytes presented in figure 3b is not commented in the text.

We expanded the description of Fig 3 and added the **two Supplementary Figures 6,7** to simplify data complexity. (Line 196-204)

2. Figure 3 summarizes a huge amount of data and is difficult to read. Perhaps consider simplification with highlight of striking results (such as the level of pDC/mDC/neutrophils in blood or BAL) presented as histograms.

The scatterplots are included now **in the new Fig 3** as well as more focused references to the sections of the figure, and for further clarity we created **the new Supplementary Figures 6 and 7**

3. The sentence line 178 is not fully correct: the level of IL8+mDC and IN and NC IL8+monocytes in blood at w12 is lower in HTLV-1A/C infected animals than HTLV-1A. They cannot mirror that of IL8+neutrophils because this cell population is not shown at week 12. In the same line, how the authors explain that proinflammatory (expressing IL8 or TNFa) mDC and pDC are not found in BAL of HTLV-1A/C infected animals at week 12 and 21? is the inflamed lung only supported by infiltration of CD64+ neutrophil in BAL and production of CCL13 cytokine?

The text has been changed as suggested (line 196-204)

4. There is no molecular weight in the WB in sup figure 8c and 8d rending difficult to identify p8 vs p12 that is claimed to be present in THP1 but not in JK. Several bands are present in THP1 expressing p16C that are not present in JK, what is the explanation?

We added the position of the molecular weight as requested in **Fig. 1b** and **Fig. 11a,c and d**

5. The GFP signal detected in figure 5b is not commented, as the experimental setting is not precise in the text nor in the figure legend. Reader understands that cells were transduced with a lentiviral vector expressing GFP that is used to control/identify transduced cells after careful reading t-of the methods. This must be correct to ease the comprehension.

The legend of **Fig. 5b** was revised to clarify that HeLa cells used in the immunofluorescence experiments were transfected with either p12/p8-HA or p16C-HA-ires-GFP-expressing lentiviral vectors.

Reviewer #3 (Remarks to the Author):

HTLV-1C lung disease linked to p16 protein inhibition of efferocytosis in macaques
Background: HTLV-1 A is the predominant subtype whereas HTLV-1 C is rare and is known to cause respiratory failure and premature death. To better understand the different pathology associated with the 2 subtypes, Sarkis Sarkis and colleagues developed a HTLV-1A/C chimeric to recapitulate infection with HTLV-1A with the pathology of HTLV-1C in a macaque model. The authors conclude that HTLV-1A/C induced lung pathology as reported in humans and accompanied with high infiltration of pro-inflammatory innate effector cells. In contrast, HTLV-1 A induced the resolution of inflammation accompanied by higher expression of IL10. Sarkis et al. identified a 16kDa fusion protein expressed by HTLV-1A/C which reduced clearance of infected T cells by macrophages/monocytes and thus providing a target for HTLV-1 C mediated lung pathology.

Overall, the conclusions of the authors are not fully supported by the data or the presentation of the data. In detail:

Major comments:

1. Is there any evidence that the protein P16C is expressed in humans and is associated with lung pathology?

No, there is no direct evidence at the moment of protein expression in humans. Neither antibodies to p16 nor tissues from lung of infected individuals are available at present. However, we found more frequent expression of the *rex-orf-I* mRNA in the lung of animals infected with HTLV-1A/C (Fig. 6c), compared with those infected with HTLV-1A. (Line 372-381

2. It is unclear whether HTLV-1A and HTLV1A/C show the same infectivity and virus replication in vivo. The two virus types were not compared head-to head in the same experiment. It is hard to judge if the statement “Infectivity of HTLV-1A and chimeric HTLV-1A/C was equivalent in macaques” is correct given this major limitation. The viral scores and relative units in Figure 2 are difficult for a reader to understand and evaluate. Viral titers in serum, PBMCs and potentially other compartments such as the BAL should have been reported with time kinetics and clinical scores.

As also specified in our response to Reviewer 1 (point 3) we recognize the importance of this point. Indeed, we have performed the Study 2 to address this and we have added the results to the manuscript (Supplementary Fig. 3e-h). We combined the results of Studies 1 and 2 and reanalyzed data on virus score, cytokines, and cellular responses in blood and lung together on 5 triple depleted, and 4 replete animals infected with HTLV-1A/C_{ol-L} (total n=9), data from 5 historical and 5 new triple depleted animals infected with HTLV-1A (Supplementary Fig. 3i). We combined the data from animals infected with HTLV-1A (total n=10) since they had a similar systemic virus score (data not shown), that remained similar to that of HTLV-1A/C_{ol-L} infected macaques (Fig. 2 d,e). Interestingly, however, the virus score in lung differed between HTLV-1A and HTLV-1AC infected animals, as did lung pathology (Fig. 6 and Supplementary Table 2).

3. Figure 3 is very complex and not well explained. No gating strategy for flow cytometry (only as text).

We added **Supplementary Fig. 5**, describing the gating strategy.

For further clarification of Fig 3 data, we have amended it to include scatterplots rather than heatmaps and provide **the new Supplementary Figures 6 and 7** for additional visualization of the data.

What was the frequency of IL10+ or TNFa+ cells straight ex vivo? Was the frequency robust enough for downstream analysis? For example: How many events of pDCs expressing IL10 were collected which allow robust analysis of such sub-populations?

The number of events analyzed in each FACS assay is provided in Raw Data table, as requested.

FACS plots of MPO staining including control staining?

We do not run a simultaneous Ig control. However, an example of how we measured MPO expression in isolated neutrophil versus other populations is shown below.

No tracking of individual cell populations in total numbers and frequencies side-by-side in order to judge whether innate immune cell numbers in BAL or blood are truly and significantly altered between the virus variants.

We did not have sufficient cells to compare the absolute number of each subset cells in BAL.

Why was week 5 chosen as the first time point?

The week 5 timepoint was chosen because all animals had seroconverted by this time.

4. Given that in this model, 3 major cell populations (CD8 T cells, NK cells and monocytes) are depleted for some period of time it is difficult to exclude that this has an impact on subsequent cell dynamics and cytokine production of all other cell types.

Further, different or heterogeneous cell discovery rates between animals and groups might (for example Suppl Figure 4b and c) make these dynamics even more complex. This is a limitation of the immunological investigations at later time points as reported in Figure 3 and 4.

We demonstrated a recovery to baseline frequency of all cell populations except CD8⁺ T-cells (**presented in the new supplementary Fig 4**)

The impact of triple depletion on immune responses and cytokine/chemokine profiles in both blood and bronchoalveolar lavage (BAL) fluid were compared between the replete (n=4, HTLV-1A/CoI-L) and triple-depleted (n=5, HTLV-1A/CoI-L; n=10, HTLV-1A) animals at multiple time points (weeks 5, 12, and 21). While significant increases in several cytokine/chemokine levels in triple depletion vs no depletion groups were observed in the blood at an early time point (week 5 post-antibody treatment) as expected (**Supplementary Fig.8**) by later time points (12 and 21), after recovery of key immune cell populations such as monocytes (week 1) and NK cells (week 5), the levels of these markers were comparable between the triple and no depletion groups (**Supplementary Fig.8a**). In contrast, in the BAL compartment, fewer significant alterations in cytokine/chemokine levels between triple and no depleted animals were detected at earlier stages (week 5), with the majority of the differences observed, mostly reduction in triple depletion, only at the later time points (weeks 12 and 21) after cell recovery (**Supplementary Fig.8b**). Importantly, most differences in cytokine/chemokine levels in BAL between triple and non-depleted animals were unique to either the HTLV-1A/CoI-L triple depleted group or the HTLV-1A triple depleted group, relative to the HTLV-1A/CoI-L no depletion group. This suggests that virus type prevails as the primary differentiating variable in the cytokine/chemokine profiles (lines 236-251)

5. The lung of HTLV-1A was not investigated for lung pathology and lung pathology for the HTLV-1A/C infected animals was not quantified.

We have performed Study 2 to address this question. Quantitation of lesion in all lobes by histology is qualitative rather than quantitative.

Only representative images are shown. Thus, the statement: “suggesting causal association between HTLV-1A/CoI-L infection and lung disease” is to some extent overinterpretation or speculation.

We appreciate the reviewer’s comment and have restrained the wording to instead say these findings raise hypotheses.

p16C staining is not shown in Figure 6.

Unfortunately, there are no antibodies available to detect p16 expression at present.

It is not clear whether this is highly expressed in the lung of HTLV1-A/C infected animals.

We found more frequent expression of the *rex-orf-I mRNA* in the lung of HTLV-1A/C than HTLV1-A infected animals (**Fig. 6c**).

No comparison to HTLV-1A animals is available to judge whether CD3 T cell clearance is prevented in HTLV-1A/C infected animals.

In the revised version of the paper, we have added an additional experiment (Study 2, n=5) of animals infected with HTLV-1A. In blood we observed no differences in virus score, but the lung of HTLV-1A/C infected animals had a higher virus burden. The hypothesis is that less cell clearance occurs in HTLV-1A/C infection of lungs.

Minor:

1. High prevalence of HTLV-1C in Aboriginal populations might point towards a human genetic component? Please discuss

We agree the genetics of the Aboriginal populations may play a role, and the historical isolation of these populations, as well as initiation rites with exposure to blood could also be underlining reasons. Our results also support the possibility of genetic difference between HTLV-1A and HTLV-1C contributing to the high prevalence.

1G: Difference in virus replication at week 1 post infection in vitro significant? This earlier time point is missing for 1H and 1I. Why did the authors look only at time points 2 weeks post co-culture?

Unfortunately, the supernatants were not collected at week 1 post co-cultivation of the irradiated 729 B cells line with the CD4+ T cells isolated from either human or macaque PBMCs.

3. Figure 5c-f: Efferocytosis rates appear overall quite low. Did the authors try to increase the ratio of Jurkat or CD4 T cells to monocytes to increase the overall efferocytosis rate?

We have not tested different ratio of cells for this paper. However, separate work is ongoing to study mechanistically how p16 inhibits efferocytosis.

Response to reviewer 3

Reviewer #3 (Remarks to the Author):

The revised manuscript by Sarkis et al. provides additional data, analysis and conclusions. The authors responded to all questions/concerns raised by the reviewer. The study was truly improved by an additional study/data for the HTLV-1A group for comparison. However, the comparison of HTLV-1A to HTLV-1A/C appears to show only moderate - small difference in the NHPs between HTLV-1A and HTLV-1A/C as presented.

We were pleased to observe that even after increasing the number of animals infected with HTLV-1A, significant differences remained between the two groups infected with either HTLV-1A or HTLV-1A/C, both at the cellular level and the cytokine/chemokine production. We did not wish to make the case that the differences were major, rather that they were significant for the animals we studied. The data suggest that even subtle differences in cytokine/chemokine production can result in different biological outcomes.

Major differences but overall, in case the authors would argue that major differences exist, they are not well visualized or analyzed. I focused primarily on the analysis in Figure 3 accompanied by Suppl Figures and raw data tables provided for the revised manuscript. In general, it was very difficult to understand the plots, analysis and how this was connected to the conclusions in the text. It took several hours to get some good idea about the nature of the data, the legends, labeling, analysis and interpretation. Details of my major comments are below:

We have further attempted to clarify Figure 3 by adding an independent range for the frequency x-axis for each cell population rather than a shared standard 0-100 range for all frequencies. This allows better distribution and visualization of the data points for each cell population. We also extended the colored alluvials under these individual axes to clearly assign which axes belongs to which plot. Additionally, we added boxplots with median and quartiles as well as asterisks for p-value levels to aid in highlighting the most significantly different cell populations. Finally, we added a new Supplementary Figure 7 that contains simply the standard beeswarm boxplots from this figure, for those who prefer a more tabular layout. We thank the reviewer for expressing their concerns and hope these modifications enable a smoother interpretation of the data.

- The gating strategy in Suppl Figure 5 appears incomplete. Is this PBMCs or BAL? Which Animal and time point? In the plot titled “Non-T, B and NK cells: Why do most cells appear as CD8+CD3+CD20-A700? Where are the CD4 T cells and B cells? Gated out before? Its unclear.

We have updated the Supplementary Figure 5 to show the gating strategy used to identify the different cell populations from both blood and BAL of the same animal. The figure was generated using data from an animal infected with the HTLV-1A virus (17P039) at week 21 post viral inoculation.

- All the CD4⁺ and CD8⁺ T cells express CD3 marker, thus CD3 excludes all the T cells.

- In macaques, NK cells express CD8 thus, the CD8 marker was used to exclude also CD8+ NK cells.
- CD20 was used to exclude B cells.
- In case this is a blood sample: In the gate Macrophage/monocytes: The majority of cells appear to be macrophages. They should not really exist at all in blood (or below 0.1%). Macrophages reside / differentiate in tissue.

The sample used for the original Supplementary Figure 5 gating was a BAL sample, but we have remade the figure as indicated above to illustrate both BAL and blood side by side, where the blood contains very low macrophage signal (0.23%).

- Having said that, the frequency of macrophages appears very low in the BAL (here the frequencies are generally reported with 80-90%). Whether this is an effect of the triple depletion is not clear. However, it does not seem to be different 21 weeks post infection or at the earlier time points (raw data table).

In this study, the frequencies of cell populations were taken in CD45⁺ cells, thus, the frequencies in the figures are low, as reported and expected. As you can see in the gating strategy (Supplementary Figure 5), out of the parent population the frequency of macrophages in BAL is 83.5%, which aligns with the suggested general range.

- Classical monocytes appear to be at 28.9% (Suppl gating strategy). This seems low considering they should make the vast majority of all monocytes (in humans).

In our current study, in blood, classical monocyte frequencies range from 51-93% across both groups at 3 timepoints (week5, 12 and 21), with mean & median at 71% for both HTLV-1A and HTLV-1A/C. In BAL, classical monocyte frequencies range from 0.7-63% across both groups at 3 timepoints, with mean 23% and median 21%. As shown in Figure 3, the classical monocyte frequencies in BAL only differ between groups at week 21, when the HTLV-1A group has median 13% and the HTLV-1A/C has median 45%. Thus, in our dataset, in blood the classical monocytes are the majority, while in BAL they are not. This can also be seen in the updated Supplementary Figure 5 where both a blood and BAL sample have been gated, with 62.9% Classical in blood and 7.8% classical in BAL.

- The ref 67 provided in the method section for NHPs monocyte subpopulations does not provide this information. In fact, in that paper I did not find any analysis of monocytes.

We have updated the reference 67 (64 in the new manuscript) ¹.

- Given my concern regarding the macrophage population, it is not clear from the information provided that the gating strategy identifies the majority of myeloid cells correctly.

The gating provided in the previous version of the manuscript was done on a BAL sample, where most of the CD3⁻CD20⁻CD8⁻CD11b⁺HLA-DR⁺ cells were macrophages. When we

applied the same gating on blood, the majority of the cells were monocytes. We have provided an updated Supplementary Figure 5 showing the gating of Blood and BAL from an HTLV-1A infected animal side-by-side, and hope this will satisfy the reviews concern.

- Further the gating for mDCs does not appear to show a distinct population.

We have updated the Supplementary Figure 5, showing gating of cell population from the same animal in blood and BAL, which clarify the gating of mDC.

- In line with this it is surprising that 14% of mDCs appear to be MPO+ (reviewer comments) considering it is a bona fide neutrophil marker.

In wild-type SPF-housed mice, MPO⁺ monocytes/macrophages were detected in peripheral blood, spleen, small and large intestines, and mesenteric lymphnodes ². MPO⁻ myeloid progenitors develop Langerhans-type dendritic cell (LC) and granulomonocytic (GM) lineages, while MPO⁺ progeny demonstrated potential to differentiate into myeloid dendritic cells (mDCs) ³. Furthermore, it has been shown that in inflammatory circumstances and under specific stimuli, monocytes derived DC can transiently express MPO during early stages of differentiation from monocytes⁴. These findings indicate that MPO expression is not limited to neutrophils but is also present in other myeloid populations. However, our study did not specifically investigate MPO expression in mDCs, only MPO on neutrophils.

- There are no flow plots showing the staining/gating for IL8, TNFa and IL10. The frequency of myeloid cells expressing these chemokine/cytokines straight ex vivo appears very high (up to 99%).

We have updated the Supplementary Figure 5, showing the gating for IL8, TNFa and IL10.

- Thus, the conclusions from this analysis appear not convincing at all. Looking at the tables provided as raw data, the gating strategy and the missing information all together it appears the differences described in the text (line 185-212) are not supported by the data and the analysis.

We hope that the additional clarifications in the gating strategy (Supplementary Figure 5), text and Figure 3 will address the reviewer's doubts.

- For the proteomics analysis of chemokines and cytokines the analysis as provided does not allow to judge whether there are indeed differences. A table with the median of each analyte in each group and an appropriate statistical test (non-parametric corrected for multiple comparisons) should be provided along with boxplots or bar graphs for each analyte. This would help the reader to fully appreciate any differences.

We have provided a new additional Supplementary Figure 8 with the requested boxplots, as well as included the suggested median table in the plotted values table for Supplementary Figure 8 for further clarity. All pairwise between-group comparisons were subjected to the non-

parametric Mann-Whitney test as reported. This is an exploratory study with a small sample size testing many non-independent variables. Because all the conclusions we draw are hypotheses that will need to be verified by subsequent studies, we want to maximize our limited power and minimize our type II error (i.e. not miss any potential hypotheses), knowing that some of our hypotheses may turn out to be false positives. Adjusting for multiple comparison would reduce our false positives, but also increase our false negatives, and since our study aim of finding the differences between HTLV-1A and HTLV-1AC infection tests numerous measures that may be quite related (i.e. non-independent), there is no clear way of determining our false positive or false negative rate beyond that of each individual measure⁵. Further, we report all the measures tested along with their unadjusted p-values, and clearly state this in the methods section, so that the reader may assess for themselves the exploratory nature of the study.

- The differences in pathology and viral load in the lung appear minor.

The detection viral mRNA was significantly different in infections by the two viruses. The difference in pathology was noted as no bronchiectasis were observed in HTLV-1A infected animals. We did not conclude that HTLV-1A does not cause lung disease in macaques, rather that HTLV-1AC does do it in a short term.

References:

1. Kapellos, T.S. *et al.* Human Monocyte Subsets and Phenotypes in Major Chronic Inflammatory Diseases. *Front Immunol* **10**, 2035 (2019).
2. Gurski, C.J. & Dittel, B.N. Myeloperoxidase as a Marker to Differentiate Mouse Monocyte/Macrophage Subsets. *Int J Mol Sci* **23** (2022).
3. Scholz, W. *et al.* Initial human myeloid/dendritic cell progenitors identified by absence of myeloperoxidase protein expression. *Exp Hematol* **32**, 270-276 (2004).
4. Palucka, K.A., Taquet, N., Sanchez-Chapuis, F. & Gluckman, J.C. Dendritic cells as the terminal stage of monocyte differentiation. *J Immunol* **160**, 4587-4595 (1998).
5. Althouse, A.D. Adjust for Multiple Comparisons? It's Not That Simple. *Ann Thorac Surg* **101**, 1644-1645 (2016).

Response to Reviewer #3:

Thank you very much for clarifying the gating strategy. I still struggle with some parts of it. In blood the majority of all CD45⁺ cells are lymphocytes. When you gate out CD3⁺ and CD20⁺ lymphocytes it appears that 80% (of all viable CD45⁺ cells) remained in the negative gate. This raises a concern about full exclusion of lymphocytes at this point.

We agree with the Reviewer that lymphocytes account for the majority of the CD45⁺ leukocytes cells in PBMCs. For additional clarifications, our assays were performed on whole blood, where neutrophils and other myeloid leukocytes rather than lymphocytes are known to constitute the major component of the CD45⁺ (1-3). This distinction (that we further highlighted) significantly impacts the observed frequency of CD3⁺CD20⁻ cells.

Vise versa in BAL it appears that 46.8% of all viable CD45⁺ are non-lymphocytes (not CD3⁺ or CD20⁺). In BAL the alveolar macrophages are large and have high autofluorescence. In the provided dotplot it appears that the CD3⁺CD20⁺ cells are higher in the FSC-H which raises the concern that auto fluorescent FSC high macrophages are gated out. If you calculate the Freq of Bal macs of all viable CD45⁺ cells this appears low as mentioned before.

We thank the reviewer for bringing this concern to our attention. Based on this suggestion, we re-analyzed the data to ensure greater accuracy and clarity. In the revised analysis of the BAL samples (from all animals at all time points) we first gated on CD45⁺ CD206⁺ CD163⁺ CD11b^{intermediate} cells (New Supplementary Fig. 5) which allowed us to accurately identify the alveolar macrophages and subsequently minimize any potential contamination from the reported cell populations. Due to the re-gating in the BAL, a small portion of results have changed, and were incorporated into the manuscript. (Fig. 3; Fig. 4d-g; Supplementary Fig. 5; Supplementary Fig. 6; Supplementary Fig. 7). The overall conclusions were not affected by the exclusion of alveolar macrophages.

Thank you very much for the gating strategy. You did not address my concern regarding the high freq of IL8/TNF α + myeloid cells straight ex vivo (without further in vitro stimulation). Do you have isotype controls or other negative controls which would allow to judge whether this is not an artefact?

To address the reviewer concern regarding the control for IL-8⁺, TNF- α ⁺ or IL10⁺, we included fluorescence minus one (FMO) staining on whole blood samples from three naïve, uninfected animals, as control for our strategy. This approach allowed us to rigorously assess background staining and confirm the specificity of cytokine detection. The FMO staining demonstrated the absence of background signal, thereby validating our gating strategy and supporting the reported cytokine expression. Representative FMO staining results are included below for your review.

References:

1. Summers C, Rankin SM, Condliffe AM, Singh N, Peters AM, and Chilvers ER. Neutrophil kinetics in health and disease. *Trends Immunol.* 2010;31(8):318-24.
2. He Z, Allers C, Sugimoto C, Ahmed N, Fujioka H, Kim WK, et al. Rapid Turnover and High Production Rate of Myeloid Cells in Adult Rhesus Macaques with Compensations during Aging. *J Immunol.* 2018;200(12):4059-67.
3. Sender R, Weiss Y, Navon Y, Milo I, Azulay N, Keren L, et al. The total mass, number, and distribution of immune cells in the human body. *Proc Natl Acad Sci U S A.* 2023;120(44):e2308511120.

REVIEWERS' COMMENTS

Reviewer #3 (Remarks to the Author):

Dear authors,

Thank you very much for the clarifications. Thank you for providing the FMO data. Since the high ex vivo expression of cytokines in myeloid cells appears unusual it would be informative to include the FMOs along the gating strategy so that readers can evaluate it as well.

As suggested by the Reviewer, the FMOs were included in the Supplementary figure along the gating strategy as Supplementary Figure 5b.

ROUND 3 REVIEWER 3 ATTACHMENT:

- The gating strategy in Suppl Figure 5 appears incomplete. Is this PBMCs or BAL? Which Animal and time point? In the plot titled “Non-T, B and NK cells: Why do most cells appear as

- In case this is a blood sample: In the gate Macrophage/monocytes: The majority of cells

-

- Classical monocytes appear to be at 28.9% (Suppl gating strategy). This seems low

- The ref 67 provided in the method section for NHPs monocyte subpopulations does not

- Given my concern regarding the macrophage population, it is not clear from the

- Further the gating for mDCs does not appear to show a distinct population.

-

- +

-

- There are no flow plots showing the staining/gating for IL8, TNFa and IL10. The frequency

- Thus, the conclusions from this analysis appear not convincing at all. Looking at the tables

We hope that the additional clarifications in the gating strategy (Supplementary Figure 5), text and Figure 3 will address the reviewer's doubts.

- For the proteomics analysis of chemokines and cytokines the analysis as provided does

- The differences in pathology and viral load in the lung appear minor. The detection viral